# Randomized Subspace Methods for Optimization on Fixed-Rank Matrix Manifolds

## Abstract

This paper provides the first randomized subspace methods for optimization over fixed-rank matrix manifolds. This allows us to avoid expensive full matrix decompositions to ensure efficient exponential map computations via at most a $2 \times 2$ eigendecomposition and rank-one updates, with low storage costs. To facilitate this, we analyze the geometries of fixed-rank matrix manifolds as Riemannian quotients of convenient product manifolds. Due to the quotient structure, subspaces of interest correspond exactly to those in the *horizontal space* of said product manifolds. A tangent subspace descent scheme is then devised by decomposing the horizontal space into orthogonal subspaces. Existing instances of tangent subspace descent depend upon the selection of a subspace from a fixed collection of ones that vary smoothly over the entire manifold. In sharp contrast to these instances on other manifolds, subspaces in our scheme are not selected from any such smoothly varying collection. Instead, the randomly selected subspace at the current iterate is carefully constructed based on the past iterates and their accompanying subspace selections. Experiments for the trace regression problem demonstrate the superiority of the methods relative to full gradient methods in terms of both CPU time and iterate count.

## 1 Introduction

This paper proposes the first randomized subspace methods for solving

$$f^* := \min_{y \in M} f(y) \tag{1}$$

where $M$ is a fixed-rank matrix manifold and $f : M \to \mathbb{R}$ is a differentiable function. We specifically consider the manifolds of $m \times n$ matrices with rank $r$, $M = \mathbb{R}^{m \times n}_r$, and $n \times n$ positive semidefinite matrices with rank $r$, $M = \mathbb{S}^{n,r}_+$. Problem (1) arises in several important data science applications such as matrix completion (Vandereycken, 2013), compressed sensing (Wei et al., 2016; Luo et al., 2024), semidefinite matrix approximation (Musco & Woodruff, 2017) and trace regression (Slawski et al., 2015; Han et al., 2021).

In the Euclidean setting, coordinate descent methods are often employed for problems that are very high-dimensional. Fundamentally, these methods operate as follows: pick a random subspace, usually from a pre-determined collection such as spans of coordinate blocks; project a first-order derived search direction onto the subspace; then move in that direction. Such methods generate iterates with a low, and often dimension-free, computational complexity while enjoying the same overall iteration complexity as standard first-order methods (Beck & Tetruashvili, 2013; Nesterov, 2012). Naturally, this is the source of their widespread popularity for high-dimensional problems.

Interest in first-order methods methods for Riemannian optimization (RO) has surged over the past decade due to the frequently high-dimension of the underlying manifolds (Zhang & Sra, 2018; 2016; Criscitiello & Boumal, 2023; Chen et al., 2024). Thus the adaptation of randomized subspace methods to this setting is an enticing goal for the RO community. To the best of our knowledge the first coordinate descent type method for RO dates back to the randomized subspace method proposed by Shalit & Chechik (2014) which applies only to the orthogonal group. The first generic framework for constructing convergent coordinate descent methods on any complete Riemannian manifold, called *Tangent Subspace Descent* (TSD), was proposed by Gutman & Ho-Nguyen (2023). A fruit

of this framework was a new coordinate descent method that extended the orthogonal group-based scheme of Shalit & Chechik (2014) to the Stiefel manifold. Later, Darmwal & Rajawat (2023) proposed a randomized subspace method, readily seen as an instance of TSD, on the manifold of (full rank) positive definite matrices. Other recent papers have focused on retraction-based randomized subspace methods for minimization over product manifolds (Li et al., 2023; Peng & Vidal, 2023; Firouzehtarash & Hosseini, 2021). Recently Han et al. (2024) devise a straightforward, but partial generalization of tangent subspace descent to incorporate retractions and vector transports. This includes a new tangent subspace descent-type scheme for optimization over the low-rank positive semi-definite (PSD) matrix manifolds. This scheme suffers from two key issues. First, the chosen retraction is not defined over the entire tangent bundle. Thus, it may lead to ill-defined iterates. Second, the subspace selection mechanism depends upon the choice of representative, implying the scheme is not well-defined on a low-rank PSD matrix manifold. By contrast, our scheme depends upon the exponential map on defined by complete quotient geometry, and thus bypasses the first issue. Additionally, our subspaces selections are independent of the representative except on a set of measure zero, which the algorithm is unlikely to ever encounter.

Our proposed randomized subspace methods for fixed-rank matrix manifolds differ from all of the aforementioned instances along two key technical directions. First (*non-smooth subspace selection*), the set of subspaces considered for selection at each iteration do not arise from any smoothly varying subspace collection. In a sense which we make rigorous in Section 3, we provide the first non-smooth method of subspace selection. Second (*lifting by Riemannian quotients*), our methods depend upon a "lift" of the problem (1) to a related convenient Riemannian product manifold via Riemannian quotient operations. These products have simple expressions for various Riemannian constructs including Riemannian metrics and exponential maps.

Combined, these two innovations permit us to construct randomized subspaces methods for fixed-rank matrices which are efficient in terms of computational time and storage, but still guarantee standard rates of convergence for first-order RO.

This paper is organized as follows. In Section 2, we provide the notation and language of RO and quotient manifolds necessary to understand all of our results in the sequel sections. In Section 3, we describe the Tangent Subspace Descent (TSD) framework and the conditions guaranteeing its convergence in the presence of Riemannian quotient structures. In particular, we describe the three necessary ingredients: a quotient geometry, a horizontal decomposition, and cheap, simple expressions for gradient projections and exponential maps. Each of our main convergence pivots on the verification of these conditions and the complexity results depend on the latter cheap expressions. We also clarify the sense in which all previous instances of TSD depend upon "smooth" subspace selections. In Section 4, we elaborate each of the three ingredients necessary for building our randomized subspace descent method on the manifold of fixed-rank matrices, $\mathbb{R}_r^{m \times n}$, via TSD. In Section 5, we elaborate each of the three ingredients necessary for building our randomized subspace descent method on the manifold of fixed-rank positive semidefinite matrices, $\mathbb{S}_+^{n,r}$, via TSD. In Section 6, we present proof-of-concept experiments that compare the Riemannian gradient descent method against our scheme in the context of trace regression over the manifold of fixed-rank positive semidefinite matrices. These experiments indicate the computational superiority of our scheme over full gradient methods.

## 2 PRELIMINARIES: RIEMANNIAN OPTIMIZATION

In this section, we describe all of the rudiments of Riemannian optimization and quotient manifolds required for our work. For full details we refer to the books of Boumal (2023, particularly Ch. 9) and Gallier & Quaintance (2020, particularly Chs. 5 and 23).

If $M$ is a Riemannian manifold, then we let $T_x M$ denote the tangent space and $\langle \cdot, \cdot \rangle_x$ the Riemannian metric of $M$ at $x \in M$. In the presence of a Riemannian metric on $M$, we can utilize Riemannian constructs such as the (Riemannian) exponential map at $x \in M$, $\text{Exp}_x : T_x M \to M$, and the Riemannian gradient, $\nabla f$, of a differentiable function $f : M \to \mathbb{R}$.

Tangent spaces, Riemannian metrics, and Riemannian exponential maps interact well with product structures. If $M_1, \ldots, M_k$ are Riemannian manifolds, then the Cartesian product $M := \prod_{i=1}^k M_i$ is a manifold whose tangent space at $x = (x_1, \ldots, x_k) \in M$ is $T_x M = \prod_{i=1}^k T_{x_i} M_i$. We can endow

the product manifold $M$ with a Riemannian metric,

$$\langle \cdot, \cdot \rangle_{x_1}^{(1)} \oplus \ldots \oplus \langle \cdot, \cdot \rangle_{x_k}^{(k)},$$

which is expressed for arbitrary $x \in M$ and $v := (v_1, \ldots, v_k), w := (w_1, \ldots, w_k) \in T_x M$ as,

$$\sum_{i=1}^{k} \langle v_i, w_i \rangle_{x_i}^{(i)}$$

where $\langle \cdot, \cdot \rangle_{x_i}^{(i)}$ is the Riemannian metric on $M_i$ at $x_i$. As Riemannian exponential maps respect product metrics, $M$'s exponential map is given as,

$$\mathrm{Exp}_x^M(v) = \left( \mathrm{Exp}_{x_1}^1(v_1), \ldots, \mathrm{Exp}_{x_k}^1(v_k) \right),$$

where $\mathrm{Exp}_{x_i}^i : T_{x_i} M_i \to M_i$ is the exponential map for $M_i$ at $x_i \in M_i$.

The basic building blocks of the product manifolds we consider in this paper are Stiefel manifolds and the manifold of positive definite matrices. The $n \times r$ Stiefel manifold is

$$\mathrm{St}(n, r) := \left\{ X \in \mathbb{R}^{n \times r} : X^\top X = I_r \right\}$$

where $I_r$ is the $r \times r$ identity matrix and $\mathbb{R}^{n \times r}$ is the set of $n \times r$ matrices. We let $\mathbb{S}_{++}^r$ denote the manifold of $r \times r$ positive definite matrices. The tangent spaces for these manifolds at arbitrary points $U \in \mathrm{St}(n, r)$ and $P \in \mathbb{S}_{++}^r$ are

$$T_U \mathrm{St}(n, r) = \left\{ UA + U^\perp B : A \in \mathrm{Skew}(r), B \in \mathbb{R}^{(n-r) \times r} \right\}, \quad T_P \mathbb{S}_{++}^r = \mathbb{S}^r \tag{2}$$

where $U^\perp \in \mathrm{St}(n, n-r)$ satisfies $U^\top U^\perp = \mathbf{0}$, and $\mathrm{Skew}(r)$ and $\mathbb{S}^r$ respectively denote the sets of $r \times r$ skew-symmetric and symmetric matrices. For Riemannian metrics, we will focus on the canonical metric on $\mathrm{St}(n, r)$ and the affine-invariant metric on $\mathbb{S}_{++}^r$, which are given by

$$\left\langle UA_1 + U^\perp B_1, UA_2 + U^\perp B_2 \right\rangle_U^{\mathrm{St}(n,r)} = \frac{1}{2} \langle A_1, A_2 \rangle + \langle B_1, B_2 \rangle$$

$$\langle D_1, D_2 \rangle_P^{\mathbb{S}_{++}^r} = \langle P^{-1/2} D_1 P^{-1/2}, P^{-1/2} D_2 P^{-1/2} \rangle, \tag{3}$$

where $\langle \cdot, \cdot \rangle$ denotes the usual Euclidean inner product between two equal-sized matrices $\langle X, Y \rangle := \mathrm{Tr}(X^\top Y)$. Closed-form formulas for the Riemannian exponential maps for these metrics (provided in Appendix A) respectively date back to Edelman et al. (1998) and Moakher (2005). These formulas are key to the construction of cheap and easy updates.

Riemannian quotient structures play a pivotal role in our methods. For each fixed-rank matrix manifold we consider, we "lift" the problem (1) to a convenient product of Stiefel and positive definite manifolds by way of Riemannian quotient operations. The convenience of such lifts derives from the simple and often efficient expressions they give for Riemannian constructs, such as exponential maps and parallel transports, on the lifted space. As the reader will soon see, the simplicity of the exponential map via lifting is of particular importance for our presented methods. A Riemannian manifold $M$ is said to be a *Riemannian quotient manifold* of a Riemannian manifold $\widetilde{M}$ if there is a surjective differentiable map $\pi : \widetilde{M} \to M$ such that $d\pi_x : [\ker(d\pi_x)]^\perp \to T_{\pi(x)} M$ is a linear isometry for each $x \in \widetilde{M}$, where $[\ker(d\pi_x)]^\perp$ is the orthogonal complement of $\ker(d\pi_x)$ in $\widetilde{M}$'s Riemannian metric. In this case, we call $\mathcal{H}_x := [\ker(d\pi_x)]^\perp$ and $\mathcal{V}_x := \ker(d\pi_x)$ the *horizontal space* and *vertical space* at $x$. Furthermore, we call $\widetilde{M}$ the *total space* and $M$ the *base space*.

In the presence of a Riemannian quotient structure, we can solve (1) by instead solving the equivalent "lifted" problem

$$\widetilde{f}^* = \min_{x \in \widetilde{M}} \widetilde{f}(x), \quad \text{where} \quad \widetilde{f}(x) := f(\pi(x)), \tag{4}$$

over the total space, $\widetilde{M}$. In fact, Boumal (2023, Prop. 9.6) states that global minimizers, local minimizers, and first/second-order critical points of (1) and (4) are in direct correspondence with each other. Moreover, Riemannian gradient methods are readily adapted to solving (4), because of the following practical formula (Boumal, 2023, Prop 9.39), for lifting Riemannian gradients on $M$ to $\widetilde{M}$ at any $x \in \widetilde{M}$:

$$\nabla \widetilde{f}(x) = d\pi_x|_{\mathcal{H}_x}^{-1} \left[ \nabla f(\pi(x)) \right] \in \mathcal{H}_x. \tag{5}$$

The quotient structure of the fixed-rank matrix manifolds will be described in Sections 4 and 5.

---

**Algorithm 1** Randomized TSD

---

**Input:** Initial point $x_0 \in M$, stepsize sequence $\{\gamma_t\}_{t \geq 0}$
**Output:** Sequence $\{x_t\}_{t \geq 0} \subset \widetilde{M}$.
**for** $t = 0, 1, 2, \ldots$ **do**
    Get subspace decomposition $S_{1,t}, \ldots, S_{m,t} \subseteq T_{x_t} M$ and distribution $P_t$ over $[m]$.
    Draw $i_t \in [m]$ randomly from $P_t$.
    Compute $x_{t+1} := \mathrm{Exp}_{x_t}\left(-\eta_t \, \mathrm{Proj}_{S_{i_t,t}}\left(\nabla f(x_t)\right)\right)$.
**end for**

---

## 3   Tangent Subspace Descent via Non-smooth Subspaces and Riemannian Quotient Manifolds

The construction of our randomized subspace methods depends upon the Tangent Subspace Descent (TSD) framework of Gutman & Ho-Nguyen (2023), which generalizes coordinate descent methods to solve problems of the form (1). In this section, we introduce the TSD framework, and expound on how our fixed-rank matrix manifold algorithms differ from all other existent TSD instances. We pay special attention to the conditions ensuring convergence and cheap iterations in the presense of Riemannian quotient structures, and thus support our developments for fixed-rank matrix manifolds.

Algorithm 1 formally elaborates the randomized TSD framework. The fundamental insight underlying this framework is that coordinate blocks in the Euclidean setting correspond to tangent subspaces in the general Riemannian setting. Thus, the algorithm operates by randomly selecting a subspace from an orthogonal collection, called a *subspace decomposition*, that spans the entire tangent space at the current iterate, projecting the negative Riemannian gradient onto this subspace, then moving in the projected direction along the manifold via the exponential map.

A *randomized subspace selection rule* provides a method of selecting a subspace decomposition at each point, together with a distribution over these subspaces. Good selection rules ensure that TSD converges and that Riemannian operations such as gradient projections and exponential map evaluations are efficiently computable.

Our methods for fixed-rank matrices differ from previous instances of TSD in two critical ways. First, the previous randomized subspace selection rules on Euclidean space, the Stiefel manifold and the positive definite manifold arise from subspaces that vary smoothly across the entire manifold. Consider a collection of tangent subspaces $S := \{S_x \subseteq T_x M\}_{x \in M}$, where each $S_x$ has dimension $k$. We say that the collection $S$ is *smoothly varying* over $M$ (or that $S$ is a *smooth distribution* (Lee, 2012, Ch. 19)) if, for any $x \in M$, there exists a set of $k$ smooth vector fields on a neighborhood $U$ of $x$ whose span is $S_y$ for all $y \in U$. We say that a randomized subspace selection rule is *smooth* if there exist smoothly varying collections $S^1, \ldots, S^m$ such that at every $x \in M$ the subspace decomposition is given by $S_x^1, \ldots, S_x^m$.

**Theorem 3.1** (Proof in Appendix B). *Euclidean coordinate descent, the TSD instances of Gutman & Ho-Nguyen (2023) for Stiefel manifolds and of Darmwal & Rajawat (2023) for the positive definite matrix manifold depend upon smooth randomized subspace selection rules.*

The second dimension along which our methods differ is that they depend upon "lifting" (1) to convenient Riemannian product manifolds by way of Riemannian quotient operations, resulting in (4). The equation (5), which relates the Riemannian gradient on $M$ to that of $\widetilde{f}$ on the total space, has particularly important ramifications for our fixed-rank manifold TSD methods applied to (4). At each $x_t \in \widetilde{M}$, instead of selecting among subspaces which span all of $T_x \widetilde{M}$, we need only select from a set of orthogonal subspaces whose sum is $\mathcal{H}_x$. We call such a set a *horizontal subspace decomposition*. That said, the same condition and convergence guarantee from Gutman & Ho-Nguyen (2023) may be used to ensure convergence of our horizontal subspace selection rules.

**Assumption 3.2** ($C$-Randomized Norm). There exists $C > 0$ such that for any $t \geq 0$,

$$\sqrt{\mathbb{E}_{i \sim P_t}\left[\left\|\mathrm{Proj}_{S_{i,t}}\left(\nabla \widetilde{f}(x_t)\right)\right\|_{x_t}^2 \mid \{x_\ell\}_{\ell=0}^t\right]} \geq C \left\|\nabla \widetilde{f}(x_t)\right\|_{x_t}. \text{ Here, } \|\cdot\|_x \text{ is the norm on } T_x \widetilde{M}$$

induced by the Riemannian metric $\langle \cdot, \cdot \rangle_x$.

**Theorem 3.3** (Gutman & Ho-Nguyen (2023, Thm. 4))**.** *Suppose that Assumption 3.2 holds. In addition, suppose that the function $\widetilde{f}$ satisfies the following smoothness assumption: there exists $\widetilde{L} > 0$ such that for all $x \in \widetilde{M}$, $v \in \mathcal{H}_x$ and $i \in [m]$ we have*

$$\widetilde{f}\left(\widetilde{\mathrm{Exp}}_x\left(\mathrm{Proj}_{S_i(x)}(v)\right)\right) \leq \widetilde{f}(x) + \left\langle \nabla\widetilde{f}(x), \mathrm{Proj}_{S_i(x)}(v)\right\rangle_x + \frac{\widetilde{L}}{2}\left\|\mathrm{Proj}_{S_i(x)}(v)\right\|_x^2,$$

*where $\{S_i(x)\}$ is the decomposition of $\mathcal{H}_x$ from Assumption 3.2. Then Algorithm 1 with $S_{i,t} = S_i(x_t)$ for $i \in [m]$, $P_t = P(x_t)$ and $\eta_t = C^2/(2\widetilde{L})$ for $t \geq 0$ yields the following convergence guarantees:*

$$\lim_{t\to\infty}\mathbb{E}\left[\left\|\nabla\widetilde{f}(x_t)\right\|_{x_t}\right] = 0, \quad \min_{s\in[t]}\mathbb{E}\left[\left\|\nabla\widetilde{f}(x_{s-1})\right\|_{x_{s-1}}\right] \leq \sqrt{\frac{2\widetilde{L}(\widetilde{f}(x_0) - \widetilde{f}^*)}{C^2 t}}. \tag{6}$$

Theorem 3.3 states that convergence of the randomized TSD scheme depends on finding a horizontal subspace decomposition satisfying Assumption 3.2, thus providing a solution method for solving (1). In Sections 4 and 5 we provide quotient manifold structure for fixed-rank matrices and fixed-rank PSD matrices, as well as subspace decompositions for the horizontal spaces which satisfy Assumption 3.2.

# 4 Quotient Geometry of the Fixed-Rank Matrix Manifold for Tangent Subspace Descent

In this section, we set forth all of the elements for ensuring the convergence and low-cost iterates of tangent subspace descent on the manifold of fixed-rank matrices, $\mathbb{R}_r^{m\times n}$: a quotient geometry, a (non-smooth) horizontal subspace decomposition, and computationally and storage-wise cheap update formulas.

We commence this section with a description of the quotient geometry for $\mathbb{R}_r^{m\times n}$. The total space, along with its tangent space at an arbitrary element, choice Riemannian metric, and Riemannian quotient map are

$$\widetilde{M}_{m\times n,r} := \mathrm{St}(m,r) \times \mathbb{S}_{++}^r \times \mathrm{St}(n,r) \tag{7a}$$

$$T_{(U,P,V)}\widetilde{M}_{m\times n,r} = T_U\mathrm{St}(m,r) \oplus T_P\mathbb{S}_{++}^r \oplus T_U\mathrm{St}(n,r) \tag{7b}$$

$$\langle\cdot,\cdot\rangle := \langle\cdot,\cdot\rangle^{\mathrm{St}(m,r)} \oplus \langle\cdot,\cdot\rangle^{\mathbb{S}_{++}^r} \oplus \langle\cdot,\cdot\rangle^{\mathrm{St}(n,r)} \tag{7c}$$

$$\pi(U,P,V) := UPV^\top, \quad \text{where } (U,P,V) \in \widetilde{M}_{m\times n,r}. \tag{7d}$$

**Lemma 4.1** (Riemannian Quotient Structure for $\mathbb{R}_r^{m\times n}$)**.** *The map $\pi$ is a Riemannian quotient map when $\mathbb{R}_r^{m\times n}$ is endowed with its embedded smooth structure and suitable Riemannian metric. Additionally, the vertical space at $(U,P,V) \in \widetilde{M}_{m\times n,r}$ is given by*

$$\mathcal{V}_{(U,P,V)} = \{(UA, PA - AP, VA) : A \in \mathrm{Skew}(r)\}.$$

The proof of Lemma 4.1 is given in Appendix C.1. Observe that our metric of interest is simply the sum of known metrics on the Stiefel and positive definite manifolds, namely the canonical metrics on $\mathrm{St}(m,r)$ and $\mathrm{St}(n,r)$ and the affine-invariant metric on $\mathbb{S}_{++}^r$. Thus, per our discussion of product metrics in Section 2, the exponential map on $\widetilde{M}_{m\times n,r}$, $\widetilde{\mathrm{Exp}}$, is given by

$$\widetilde{\mathrm{Exp}}_{(U,P,V)}\left(UA + U^\perp B, D, V\tilde{A} + V^\perp\tilde{B}\right) := \left(\mathrm{Exp}_U^{\mathrm{St}(n,r)}(UA + U^\perp B), \mathrm{Exp}_P^{\mathbb{S}_{++}^r}(D),\right.$$
$$\left.\mathrm{Exp}_V^{\mathrm{St}(m,r)}(V\tilde{A} + V^\perp\tilde{B})\right).$$

## 4.1 Horizontal Subspace Decomposition

In this subsection, we describe the second of our three ingredients, a horizontal subspace decomposition. Proofs of results in this subsection are found in Appendix C.2. First, we explicitly describe the horizontal space under the metric in (7c).

**Lemma 4.2** (Horizontal Space for $\mathbb{R}_r^{m \times n}$). *For any $(U, P, V) \in \widetilde{M}_{m \times n, r}$, we have*

$$\mathcal{H}_{(U,P,V)} = \left\{ Z(D, A, B_1, B_2) : \begin{array}{l} B_1 \in \mathbb{R}^{(m-r) \times r}, \\ B_2 \in \mathbb{R}^{(n-r) \times r}, \\ D \in \mathbb{S}^r, A \in \mathrm{Skew}(r) \end{array} \right\}$$

*where*

$$Z(D, A, B_1, B_2) := \Big( U(P^{-1}D - DP^{-1} + A) + U^\perp B_1, $$
$$\frac{1}{2}D, V(P^{-1}D - DP^{-1} - A) + V^\perp B_2 \Big).$$

The above explicit description of the horizontal space enables us to describe our computationally advantageous decomposition in the next result, and we also show that Assumption 3.2 holds. We introduce some notation for convenience. If $P \in \mathbb{S}_{++}^r$ has the spectral decomposition, $P = Q \Lambda Q^\top$ where $Q = [q_1 \ \cdots \ q_r] \in O(r)$ and $\Lambda = \mathrm{Diag}(\lambda_1, \ldots, \lambda_r) \succ 0$, then denote

$$D_{ij}^P := Q \Lambda^{1/2}(e_i e_j^\top + e_j e_i^\top) \Lambda^{1/2} Q^\top = (\lambda_i \lambda_j)^{1/2}(q_i q_j^\top + q_j q_i^\top), \quad H_{ij} := q_i q_j^\top - q_j q_i^\top$$

for any $i, j \in [r]$.

**Proposition 4.3** (Horizontal Decomposition for $\mathbb{R}_r^{m \times n}$). *Let $(U, P, V) \in \widetilde{M}_{m \times n, r}$ and $P = Q \Lambda Q^\top$ be a spectral decomposition of $P$. Define the collection of horizontal subspaces*

$$S_{1,k}(U, P, V) := \left\{ (u q_k^\top, 0, 0) : u \in \ker(U^\top) \right\} \tag{8a}$$

$$S_{2,k}(U, P, V) := \left\{ (0, 0, v q_k^\top) : v \in \ker(V^\top) \right\} \tag{8b}$$

$$S_{\mathbb{S},ij}(U, P, V) := \left\{ \tau Z(D_{ij}^P, 0, 0, 0) : \tau \in \mathbb{R} \right\} \tag{8c}$$

$$S_{\mathrm{Skew},k\ell}(U, P, V) := \left\{ \tau \left( U H_{k\ell}, 0, -V H_{k\ell} \right) : \tau \in \mathbb{R} \right\}, \tag{8d}$$

*where $i, j, k, \ell \in [r]$, $k < \ell$ for $S_{\mathrm{Skew},k,\ell}(U, P, V)$, and $Z$ is as in Lemma 4.2. This forms an horizontal subspace decomposition of $\mathcal{H}_{(U,P,V)}$ under the metric (7c). Moreover, there are $m = \frac{r(2r+3)}{2}$ different subspaces in total, so the subspace selection rule that chooses uniformly from this collection satisfies Assumption 3.2 with $C = \frac{1}{\sqrt{m}} = \sqrt{\frac{2}{r(2r+3)}}$.*

As mentioned in the introduction and Section 3, this subspace selection rule is not smooth.

**Proposition 4.4** (Non-smooth Randomized Subspace Selection for $\mathbb{R}_r^{m \times n}$). *If $r > 2$, then the subspace selection rule induced by the horizontal decomposition (8) is not smooth.*

### 4.2 EFFICIENT UPDATE FORMULAS: EXPONENTIAL MAPS & GRADIENT PROJECTIONS

We now show how the horizontal decomposition (8) ensures cheap exponential map evaluations and gradient projections both in terms of computation and storage. This ensures that Algorithm 1 for $\mathbb{R}_r^{m \times n}$ with (8) can be implemented efficiently. Explicit formulas and proofs of results in this section are given in Appendix C.3.

**Theorem 4.5** (TSD Exponential Map Complexity on the $\mathbb{R}_r^{m \times n}$). *Let $(U, P, V) \in \widetilde{M}_{m \times n, r}$, $P = Q \Lambda Q^\top$ be a spectral decomposition for $P$, $W \in \mathcal{H}_{(U,P,V)}$, and $(U_+, P_+, V_+) = \widetilde{\mathrm{Exp}}_{(U,P,V)}(W)$. Given $UQ, VQ$ and $\Lambda$, we can efficiently compute a spectral decomposition $P_+ = Q_+ \Lambda_+ Q_+^\top$ for $P_+$, as well as $U_+ Q_+$ and $V_+ Q_+$ if $W$ belongs to one of the subspaces in (8). More precisely:*

1. *$W \in S_{1,k}(U, P, V)$: $U_+ Q_+$ can be obtained via a computation of a cosine, sine, and rank-one matrix update. All other components $V_+ Q_+ = VQ$, $\Lambda_+ = \Lambda$ remain unchanged.*

2. *$W \in S_{2,k}(U, P, V)$: $V_+ Q_+$ can be obtained via a computation of a cosine, sine, and rank-one matrix update. All other components $U_+ Q_+ = UQ$, and $\Lambda_+ = \Lambda$ remain unchanged.*

3. *$W \in S_{\mathbb{S},ij}(U, P, V)$ with $i < j$: $\Lambda_+$ is obtained by a $2 \times 2$ matrix diagonalization, whose entries are obtained via hyperbolic cosine and sine computations, while $U_+ Q_+$ and $V_+ Q_+$ are obtained via cosine and sine computations, and rank-one matrix updates.*

4. $W \in S_{\mathbb{S},ii}(U, P, V)$: $\Lambda_+$ can be obtained by the computation of a scalar exponential function, while $U_+Q_+ = UQ$, $V_+Q_+ = VQ$ remain unchanged.

5. $W \in S_{\mathrm{Skew},k\ell}(U, P, V)$: $U_+Q_+$ and $V_+Q_+$ can be obtained via cosine and sine computations, together with a rank-one matrix update, while $\Lambda_+ = \Lambda$ remain unchanged.

*In other words, we only need to store and update $UQ$, $VQ$ and $\Lambda$ at each iteration of Algorithm 1.*

The final component needed to implement Algorithm 1 is to describe how we can compute gradients and their projections using the stored data $UQ$, $VQ$ and $\Lambda$.

To make this precise, recall that we are given a function $f : \mathbb{R}_r^{m \times n} \to \mathbb{R}$, and we aim to solve (4), where $\widetilde{f}(U, P, V) := f(UPV^\top)$. Suppose now that $f$ can be extended naturally to some $\bar{f} : \mathbb{R}^{m \times n} \to \mathbb{R}$. Then the Riemannian gradient and projection operations can be computed efficiently.

**Proposition 4.6** (TSD Horizontal Subspace Projection Complexity on $\mathbb{R}_r^{m \times n}$). *Suppose the following matrices stored for a given $(U, P, V) \in \widetilde{M}_{m \times n, r}$: (1) a spectral decomposition $P = Q\Lambda Q^\top$ for $P$ along with $UQ$ and $VQ$; and (2) the (Euclidean) gradient, $G := \nabla \bar{f}(UPV^\top) \in \mathbb{R}^{m \times n}$, where $\bar{f} : \mathbb{R}^{m \times n} \to \mathbb{R}$ is a Euclidean extension of $f : \mathbb{R}_r^{m \times n} \to \mathbb{R}$. Projecting $\nabla \widetilde{f}(U, P, V)$ onto a subspace $H$ from (8) can be computed using data $G$, $\Lambda$, $UQ$ and $VQ$, and with operations involving matrix-vector products with $m \times n$-, $m \times r$- and $n \times r$-matrices. The total computational cost is at most $\mathcal{O}(mn)$.*

# 5 QUOTIENT GEOMETRY OF THE FIXED-RANK POSITIVE SEMIDEFINITE MATRIX MANIFOLD

In this section, we set forth all of the elements for tangent subspace descent on the manifold of fixed-rank positive semidefinite matrices, $M := \mathbb{S}_+^{n,r}$. As for the manifold of fixed-rank matrices, these include a quotient geometry, a horizontal subspace decomposition, and computationally and storage-wise cheap update formulas.

We start with a description of a quotient geometry for $\mathbb{S}_+^{n,r}$. The total space, along with its tangent space at an arbitrary element, choice Riemannian metric, and Riemannian quotient map are

$$\widetilde{M}_{n,r,+} := \mathrm{St}(m, r) \times \mathbb{S}_{++}^r, \quad T_{(U,P)}\widetilde{M}_{n,r,+} = T_U \mathrm{St}(m, r) \oplus T_P \mathbb{S}_{++}^r \tag{9a}$$

$$\langle \cdot, \cdot \rangle := \langle \cdot, \cdot \rangle^{\mathrm{St}(m,r)} \oplus \langle \cdot, \cdot \rangle^{\mathbb{S}_{++}^r} \tag{9b}$$

$$\pi(U, P) := UPU^\top, \quad \text{where } (U, P) \in \widetilde{M}_{n,r,+}. \tag{9c}$$

**Lemma 5.1** (Riemannian Quotient Structure for $\mathbb{S}_+^{n,r}$). *The map $\pi$ is a Riemannian quotient map when $\mathbb{S}_+^{n,r}$ is endowed with its embedded smooth structure and a suitable Riemannian metric. Additionally, the vertical space at $(U, P) \in \widetilde{M}_{n,r,+}$ is given by*

$$\mathcal{V}_{U,P} = \{(UA, PA - AP) : A \in \mathrm{Skew}(r)\}.$$

The proof of Lemma 5.1 is in Appendix D.1. Once again, recognizing our metric is product, we readily see the exponential map on $\widetilde{M}_{n,r,+}$, denoted $\widetilde{\mathrm{Exp}}$, is given by

$$\widetilde{\mathrm{Exp}}_{U,P}\left(UA + U^\perp B, D\right) = \left(\mathrm{Exp}_U^{\mathrm{St}(n,r)}(UA + U^\perp B), \mathrm{Exp}_P^{\mathbb{S}_{++}^{r \times r}}(D)\right).$$

## 5.1 HORIZONTAL SUBSPACE DECOMPOSITION

We now describe the horizontal decomposition; proofs of results in this subsection are in Appendix D.2. With the metric in (9b), we provide an explicit descriptions of the horizontal space.

**Lemma 5.2** (Horizontal Space of $\mathbb{S}_+^{n,r}$). *For any $(U, P) \in \widetilde{M}_{n,r,+}$, any vector in $\mathcal{H}_{(U,P)}$ can be written uniquely in the form $\left(U(P^{-1}D - DP^{-1}) + U^\perp B, \frac{1}{2}D\right)$ for some $B \in \mathbb{R}^{(n-r) \times r}$, $D \in \mathbb{S}^r$.*

We now provide a subspace decomposition for $\mathcal{H}_{(U,P)}$. Like our decomposition for the set of fixed-rank matrices, this decomposition depends upon a spectral decomposition for $P$.

**Proposition 5.3** (Horizontal Decomposition for $\mathbb{S}_+^{n,r}$). *Let $(U, P, V) \in \widetilde{M}_{n,r,+}$ and $P = Q\Lambda Q^\top$ be a spectral decomposition of $P$. The collection of horizontal subspaces*

$$S_k(U, P) := \left\{ (vq_k^\top, 0) : v \in \ker(U^\top) \right\} \tag{10a}$$

$$S_{ij}(U, P) := \mathrm{Span}\left( U(P^{-1}D_{ij}^P - D_{ij}^P P^{-1}), \frac{1}{2}D_{ij}^P \right) \tag{10b}$$

*where $1 \leq i \leq j \leq r$ and $1 \leq k \leq r$, forms an orthogonal decomposition of $T_{(U,P)}\widetilde{M}_{n,r,+}$ under the metric* (9b). *Moreover, there are $m = \frac{r(r+3)}{2}$ different subspaces in total, so the subspace selection rule that chooses uniformly from this collection satisfies Assumption 3.2 with $C = \frac{1}{\sqrt{m}} = \sqrt{\frac{2}{r(r+3)}}$.*

As for the fixed-rank matrix manifold, and as foreshadowed in the introduction as well as Section 3, this randomized subspace selection rule is not smooth.

**Proposition 5.4** (Non-smooth Randomized Subspace Selection for $\mathbb{S}_+^{n,r}$). *If $r > 2$, then the subspace selection rule induced by the horizontal decomposition in Proposition 5.3 is not smooth.*

## 5.2 Efficient Update Formulas: Exponential Maps & Gradient Projections

The horizontal decomposition (10) ensures cheap exponential map evaluations and gradient projections both in terms of computation and storage. This ensures that Algorithm 1 for $\mathbb{S}_+^{n,r}$ with (10) can be implemented efficiently. Explicit formulas of results in this section are given in Appendix D.3. The proofs largely mimic those of results in Section 4.2, we provide proof outlines in Appendix D.3.

**Theorem 5.5** (TSD Exponential Map Complexity on $\mathbb{S}_+^{n,r}$). *Let $(U, P, V) \in \widetilde{M}_{n,r,+}$, $P = Q\Lambda Q^\top$ be a spectral decomposition for $P$, $W \in \mathcal{H}_{(U,P)}$, and $(U_+, P_+) = \widetilde{\mathrm{Exp}}_{(U,P)}(W)$. Given $UQ$ and $\Lambda$, we can efficiently compute a spectral decomposition $P_+ = Q_+\Lambda_+Q_+^\top$ for $P_+$, as well as $U_+Q_+$ if $W$ belongs to one of the subspaces in (10). More precisely (ignoring the $V$ components):*

1. *$W \in S_k(U, P)$: The update to $U_+Q_+$ is identical to case 1 of Theorem 4.5.*

2. *$W \in S_{\mathbb{S},ij}(U, P)$ with $i < j$: The update to $U_+Q_+, \Lambda_+$ is identical to case 3 of Theorem 4.5.*

3. *$W \in S_{\mathbb{S},ii}(U, P)$: The update to $\Lambda_+$ is identical to case 4 of Theorem 4.5.*

*In other words, we only need to store and update $UQ$ and $\Lambda$ at each iteration of Algorithm 1.*

**Proposition 5.6** (TSD Horizontal Subspace Projection Complexity on $\mathbb{S}_+^{n,r}$). *Suppose the following matrices are stored for a given $(U, P) \in \widetilde{M}_{n,r,+}$: (1) a spectral decomposition $P = Q\Lambda Q^\top$ for $P$ along with $UQ$ and $VQ$; and (2) the (Euclidean) gradient, $G := \nabla \bar{f}(UPV^\top) \in \mathbb{R}^{n \times n}$, where $\bar{f} : \mathbb{R}^{n \times n} \to \mathbb{R}$ is a Euclidean extension of $f : \mathbb{S}_+^{n,r} \to \mathbb{R}$. Projecting $\nabla \widetilde{f}(U, P, V)$ onto a subspace $H$ from (10) can be computed using data $G$, $\Lambda$, and $UQ$, and operations involving matrix-vector multiplications with $n \times n$- and $n \times r$-matrices. The total computational cost is $\mathcal{O}(n^2)$.*

## 6 Numerical Study

We perform a proof-of-concept numerical study to explore the efficacy of TSD versus Riemannian gradient descent (RGD) on the problem of trace regression over PSD matrices (Slawski et al., 2015; Han et al., 2021). Specifically, we will solve the following problem: $\min_{(U,P)\in\widetilde{M}_{n,r,+}} \frac{1}{2N}\sum_{p\in[N]}(y_p - x_p^\top UPU^\top x_p)^2$. The data for this problem are pairs $(x_p, y_p) \in \mathbb{R}^n \times \mathbb{R}$ for $p \in [N]$. For our study, we will use simulated data generated from the following model (see Han et al., 2021, Sec. 4): $y_p = x_p^\top X_* x_p + \epsilon_p$, $X_* \in \mathbb{S}_+^n$, $\epsilon_p \sim N(0, \sigma^2)$. We set $\sigma = 0.1$ $X_* = \sum_{i\in[r]} v_{i,*}v_{i,*}^\top$ where $v_{i,*} \sim N(0, I_n)$, and $N = 1000$. We set $(n, r) \in \{(10, 5), (25, 10), (50, 10), (50, 25)\}$, and generate 30 instances for each setting. For each instance, we run 2000 iterations of RGD, and 2000 "cycles" of TSD; each cycle is counted as $r(r+1)/2 + r$ iterations (this is the number of different subspaces in our decomposition). Step sizes for each algorithm are chosen via a standard backtracking Armijo line search rule. We stop the TSD algorithm

when (best RGD objective) $\geq 99.9\% \times$ (best TSD objective at time $t$) and running two more cycles. Our performance metric is the relative percentage gap: $\text{gap}_t = \frac{\text{best objective across all algorithms}}{\text{best objective at time } t} \times 100\%$. Figure 1 plots the gap against the number of cycles as well as total CPU time. We see that TSD has an advantage over RGD, but as the rank $r$ increases, this advantage diminishes slightly.

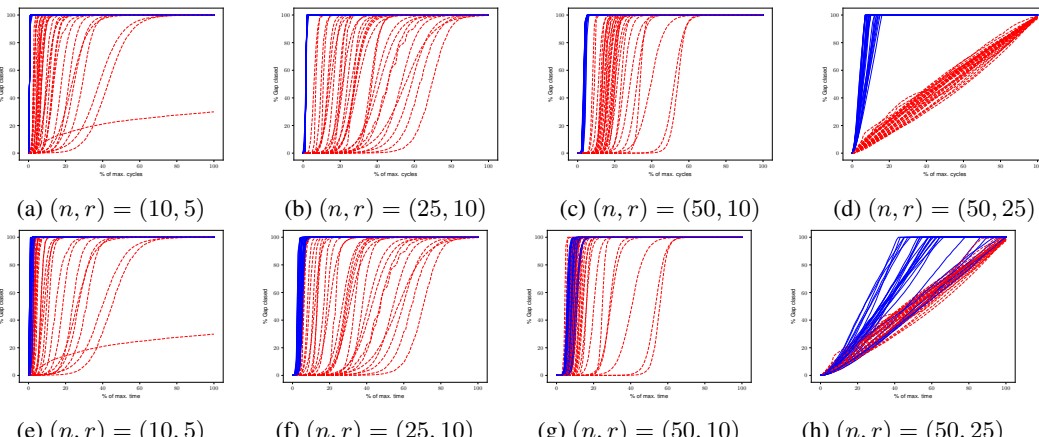

(a) $(n, r) = (10, 5)$    (b) $(n, r) = (25, 10)$    (c) $(n, r) = (50, 10)$    (d) $(n, r) = (50, 25)$

(e) $(n, r) = (10, 5)$    (f) $(n, r) = (25, 10)$    (g) $(n, r) = (50, 10)$    (h) $(n, r) = (50, 25)$

Figure 1: Plots of percentage gap closed vs number of cycles (top row) and time (in seconds; bottom row) when $N = 1000$. Red lines are RGD; blue lines are TSD. Experiments performed on standard desktop PC with 2.4GHz processor and 32GB memory.

## 6.1 COMPLEXITY OF OPERATIONS

TSD has two costly operations: exponential map and gradient projection computations. Choosing a selection rule that ensures simple exponential map computation depends upon a manifold's structure. On the other hand, choosing one that ensures simple, projected gradient computations depends upon a problem's structure. Below we describe TSD's computational benefits in the fixed-rank matrix manifold setting along both of these dimensions.

Naturally, the worst-case exponential map complexity for the (PSD) fixed-rank matrix manifold is the sum of the analogous complexities over the Stiefel and positive definite (PD) matrix manifolds. For the $n \times r$ Stiefel manifold, with its canonical metric, and the $r \times r$ PD matrix manifold with its affine-invariant metric, computing the exponential maps costs $O(nr^2)$ and $O(r^3)$ respectively (Edelman et al., 1998). By contrast, the exponential map computations along TSD's considered subspaces incur at most an $O(n)$ and $O(r)$ costs in the Stiefel and PD components respectively.

The complexity of computing projected gradients is intimately tied to a problem's structure. Without any knowledge of such structure, the computational cost equals the sum of two others: computing a (possibly total Euclidean) gradient and computing a tangent subspace projection. For the fixed-rank matrix manifolds considered here, projecting a vector from the tangent space of either the total space, or its ambient Euclidean space, requires matrix-vector products. For the $m \times n$ rank $r$ and $n \times n$ rank $r$ PSD manifolds, according to Proposition 5.6 the projection costs alone are at most $O(mr + nr + mn)$ and $O(nr + n^2)$, respectively.

That said, TSD could drastically reduce this complexity for well-structured problems. Exploiting the gradient's structure, it may be possible to very efficiently compute it only along the selected subspace, thus cheaply consolidating the gradient and projection computations. This is indeed the case for our trace regression experiments. Recalling that $N$ is the number of data points $(x_p, y_p)$ where $p \in [N]$, the complexity of the trace regression projected gradient computations can be deduced from applying Proposition 5.6. After some simplification, these require $O(N + Nr + nr)$ and $O(N)$ for the subspaces in (10) respectively. By contrast, the Riemannian gradient takes $O(Nnr + Nr^2 + Nn^2)$ operations to compute.

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

# APPENDIX MATERIAL FOR

# "Randomized Subspace Methods for Optimization on Fixed-Rank Matrix Manifolds"

This document provides all supplemental material, with exception to code, for the paper "Randomized Subspace Methods for Optimization on Fixed-Rank Matrix Manifolds".

## A  SUPPLEMENT TO SECTION 2

For Section 2, the only supplemental materials needed are the closed form expressions for the exponential maps on the Stiefel manifold, $\mathrm{St}(n,r)$, with its canonical metric, and on $\mathbb{S}^r_{++}$, with its affine-invariant metric. These maps are

$$\mathrm{Exp}^{\mathrm{St}(n,r)}_U(UA + U^\perp B) = \begin{bmatrix} U & U^\perp \end{bmatrix} \mathrm{Expm}\left( \begin{bmatrix} A & -B^\top \\ B & 0_{m-r,m-r} \end{bmatrix} \right) \begin{bmatrix} I_r \\ 0_{m-r,r} \end{bmatrix} \tag{11a}$$

$$\mathrm{Exp}^{\mathbb{S}^r_{++}}_P(D) = P^{1/2} \mathrm{Expm}(P^{-1/2} D P^{-1/2}) P^{1/2}. \tag{11b}$$

for $A \in \mathrm{Skew}(r)$, $B \in \mathbb{R}^{(n-r)\times r}$, and $D \in \mathbb{S}^r$. The formulae for the Stiefel components geodesics taken from Edelman et al. (1998), and those for positive definite manifold date back at least to Moakher (2005).

## B  SUPPLEMENT TO SECTION 3

In this supplement to Section 3, we prove that the subspace selection rules found in all previous TSD instances are non-smooth (Theorem 3.1).

First, let us recall each of the subspace decompositions for each TSD instance. Euclidean coordinate descent is the easiest to describe. Given a partition $\{B_1, \ldots, B_p\}$ of $\{1, \ldots, m\}$, the subspace decomposition for coordinate descent is

$$\left\{ \mathrm{Span}\left( \{e_i\}_{i \in B_1} \right), \ldots, \mathrm{Span}\left( \{e_i\}_{i \in B_p} \right) \right\}.$$

The subspace decomposition for the TSD instance on $\mathrm{St}(n,r)$ at $U \in \mathrm{St}(n,r)$ from Gutman & Ho-Nguyen (2023) is

$$\left\{ \mathrm{Span}(U H_{ij}) : 1 \le i < j \le p \right\} \cup \left\{ \ker(U^\top) e_i^\top : 1 \le i \le n-r \right\}$$

The subspace decomposition for the TSD instance on $\mathbb{S}^n_{++}$ at $X \in \mathbb{S}^n_{++}$ of Darmwal & Rajawat (2023) is

$$\left\{ \mathrm{Span}\left[ B_i(X) B_j(X)^\top + B_j(X) B_i(X)^\top \right] : 1 \le i < j \le n \right\}$$

where $B(X)$ is the (unique) Cholesky factor of $X$, $B_k(X)$ denotes the $k$-th column of $B(X)$.

We recall and prove our theorem regarding the smoothness of these subspace decompositions.

**Theorem 3.1.** *Euclidean coordinate descent, the TSD instances of Gutman & Ho-Nguyen (2023) for Stiefel manifolds and of Darmwal & Rajawat (2023) for the positive definite matrix manifold depend upon smooth randomized selection rules.*

*Proof of Theorem 3.1.* The manifolds $\mathbb{R}^m$, $\mathrm{St}(n,r)$, and $\mathbb{S}^n_{++}$ are Euclidean spaces or properly embedded submanifolds of a Euclidean space. If $V$ is a global extension of a (rough) vector field on an properly embedded submanifold of a Euclidean space, then we can simply regard it as a function between Euclidean spaces. Thus, smoothness of $V$ coincides with the standard smoothness of such functions. With this mind, we can prove each of the subspace decompositions is smooth. We present the proof of smoothness for each decomposition separately.

$\mathbb{R}^m$: Each subspace $\text{Span}\left(\{e_i\}_{i \in B_j}\right)$ has $\{e_i\}_{i \in B_j}$ as its basis, and each $e_i$ defines a global smooth vector field. This completes the proof of smoothness.

$\text{St}(n, r)$: We must consider two different classes of tangent subspaces $\text{Span}(U H_{ij})$ and $\ker(U^\top)e_i^\top$. First, we consider a subspace of the form $\text{Span}(U H_{ij})$. By definition, it is spanned by $U H_{ij}$ which is nothing more than the restriction of the smooth function $U \mapsto U H_{ij}$ to the $\text{St}(n, r)$.

Second, we consider a subspace of the form $\ker(U^\top)e_i^\top$. Fix a matrix $U_0 \in \text{St}(n, r)$. Let $U_0^\perp \in \text{St}(n, n - r)$ such that $U_0^\top U_0^\perp = 0$. For $1 \leq j \leq n - r$, let $v_j : \mathbb{R}^{n \times r} \to \mathbb{R}^{n \times r}$ be defined as $v_j(X) := (I - X X^\top) U_0^\perp e_j e_i^\top$. Each entry of $v_j(X)$ is quadratic in the entries of $X$, so it is smooth. By extension, its restriction to $\text{St}(n, r)$ is smooth. It remains only to show that $\{v_j(U)\}_{j=1}^{n-r}$ span $\ker(U^\top)e_i^\top$ for each $U$ in an open neighborhood of $U_0$ in $\text{St}(n, r)$. Observe for all $U \in \text{St}(n, r)$ and $1 \leq j \leq n - r$ that

$$U^\top v_j(U) = U^\top (I - U U^\top) U_0^\perp e_j e_i^\top = (U^\top - U^\top) U_0^\perp e_j e_i = 0$$

so $v_j(U) \in \ker(U^\top)e_i^\top$. Moreover, $v_j(U_0) = U_0^\perp e_j e_i^\top$ and the columns of $U_0^\perp$ form a basis for $\ker(U_0^\top)$, so $\{v_j(U_0)\}_{j=1}^{n-r}$ is basis for $\ker(U_0^\top)e_i^\top$. Thus, there must be a $(n-r) \times (n-r)$ submatrix of

$$\begin{bmatrix} v_1(U) & \dots & v_{n-r}(U) \end{bmatrix}, \tag{12}$$

which we denote $M(U)$, such that $\det[M(U_0)] \neq 0$. We must have that $\det[M(U)] \neq 0$ on some neighborhood of $U$ as $M(U)$ depends continuously on $U$, as the entries of the matrix in (12) continuously depend on $U$, and the determinant is continuous. Consequently, $\{v_j(U)\}_{j=1}^{n-r}$ spans $\ker(U^\top)e_i^\top$ for $U$ in this neighborhood.

$\mathbb{S}^n_{++}$: To show that $\text{Span}\left(B_i(X)B_j(X)^\top + B_j(X)B_i(X)^\top\right)$ is smooth, it suffices to show that each $B(X)$ smoothly depends upon $X \in \mathbb{S}^n$ since this implies each $B_i(X)$ does. To prove that $B(X)$ smoothly depends on $X$, we turn to the outer-product version of the Cholesky algorithm.

A positive definite matrix can be written in block form as

$$A = \begin{bmatrix} I_i & 0 & 0 \\ 0 & a & b^\top \\ 0 & b & B \end{bmatrix}. \tag{13}$$

where $0 \leq i \leq n$, $a > 0$, $b \in \mathbb{R}^{(n-i-1) \times 1}$, and $B \in \mathbb{R}^{(n-i-1) \times (n-i-1)}$. We adopt the convention that $I_0$ is the empty matrix. If $i = n$, then $A = I_n$. We may write $A$ as the conjugation of a positive definite matrix by a lower-triangular matrix as follows:

$$A = \begin{bmatrix} I_i & 0 & 0 \\ 0 & \sqrt{a} & 0 \\ 0 & \frac{1}{\sqrt{a}}b & I_{n-i-1} \end{bmatrix} \begin{bmatrix} I_{i+1} & 0 \\ 0 & B - \frac{1}{a}bb^T \end{bmatrix} \begin{bmatrix} I_i & 0 & 0 \\ 0 & \sqrt{a} & \frac{1}{\sqrt{a}}b^\top \\ 0 & 0 & I_{n-i-1} \end{bmatrix}$$

Let $L$ be the lower-triangular matrix, and $A_+$ be the symmetric matrix being in conjugated in the above equation. Observe that $A_+$ is not just symmetric, it is positive definite because $A$ is positive definite, $L$ is invertible, being a lower triangular matrix with positive diagonal, and $A_+ = L^{-1}AL^{-\top}$. Moreover, the entries of both $L$ and $A_+$ depend smoothly upon $A$. Applying this process inductively with $A$ initially equal to $X$, we produce a sequence of invertible lower-triangular matrices $L_1, \dots, L_n$ and positive definite matrices $A_1, \dots, A_{n+1}$ satisfying $A_i = L_i A_{i+1} L_i^\top$, and such that the entries of $L_i$ of $A_{i+1}$ depend smoothly on the entries of $A_i$. Iterating backward, we conclude $L_i$ and $A_{i+1}$ smoothly depend on $A_1 = X$. To ease notation, let $B^*(X) := L_1 L_2 \cdot L_n$. Applying the equation $A_i = L_i A_{i+1} L_i^\top$ inductively, we deduce

$$X = A_1 = B^*(X) A_{n+1} B^*(X)^\top = B^*(X) B^*(X)^\top,$$

because $A_{n+1} = I_n$ as each $A_i$ is of the form (13). The matrix $B^*(X)$ is lower-triangular, being the product of lower-triangular matrices, and satisfies $M = B^*(M)B^*(M)^\top$ so it must be the Cholesky factor of $X$, i.e. $B^*(X) = B(X)$. As each $L_i$ depends smoothly on $X$, it follows that $B(X)$ does as well. $\qquad\square$

## C  SUPPLEMENT TO SECTION 4

In this section, we use the notation $M = \mathbb{R}_r^{m \times n}$ and $\widetilde{M}_{m \times n, r} = \mathrm{St}(m, r) \times \mathbb{S}_{++}^r \times \mathrm{St}(n, r)$.

### C.1  RIEMANNIAN QUOTIENT STRUCTURE FOR THE FIXED-RANK MATRIX MANIFOLD

This subsection focuses on the proof of Lemma 4.1, which establishes the Riemannian quotient structure on the fixed-rank matrix manifold, $M = \mathbb{R}_r^{m \times n}$. The crux of our proof is the application two theorems, the Global Rank Theorem (Lee, 2012, Theorem 4.14), and (Lee, 2018, Theorem 2.28) which gives sufficient conditions for a smooth submersion to become a Riemannian quotient map. We refer the reader to (Lee, 2012, Ch. 3, 4, & 7), (Lee, 2018, Ch. 2) for the definitions of all the geometric technology employed in these theorems.

Before recalling these theorems, we recall a few definitions. A smooth map between manifolds, $F : M \to N$, is said to be *constant rank* if $\mathrm{rank}(dF_p) = \mathrm{rank}(dF_q)$ for all $p, q \in M$, and a *submersion* if $dF_p$ is surjective for all $p \in M$. A *(smooth) group action* of a Lie group $G$ on a manifold $\widetilde{M}$ is a collection of smooth maps $\tau_g : \widetilde{M} \to \widetilde{M}$ parametrized by $g \in G$ satisfying two properties. The first is that $\tau_e = \mathrm{id}_{\widetilde{M}}$, i.e., the map at the identity element $e \in G$ is simply the identity map. The second is that $\tau_{gh} = \tau_h \circ \tau_g$ for any $g, h \in G$; in other words, the map at $gh$ is obtained by first applying the map at $g$, then the map at $h$. When possible, we write $p \cdot g$ in place of $\tau_g(p)$. When $\widetilde{M}$ is a Riemannian manifold we say the action is *isometric* if $\tau_g$ is an isometry of $\widetilde{M}$ for all $g \in G$. We say it is *vertical* with respect to $F : \widetilde{M} \to N$ if $F(p \cdot g) = F(p)$ for all $g \in G$ and $p \in \widetilde{M}$, and is said to be *transitive on fibers* if for all $q, p \in \widetilde{M}$ such that $F(q) = F(p)$ there exists $g \in G$ ensuring $q = p \cdot g$.

Now, we recall the two theorems we intend to apply:

1. The Global Rank Theorem (Lee, 2012, Theorem 4.14) states that a smooth, surjective map between manifolds with constant rank is a submersion.

2. The result of Lee (2018, Theorem 2.28) states that if $\pi : \widetilde{M} \to M$ is a surjective submersion and $\widetilde{M}$ is endowed with Riemannian metric and a group action that is isometric, vertical, and transitive on fibers, then there exists a Riemannian metric on $M$ such that $\pi : \widetilde{M} \to M$ is a Riemannian quotient map.

We shall first classify the fibers of $\pi$, and exhibit a smooth group action of the $r \times r$-orthogonal group, $O(r)$, on $\widetilde{M}_{m \times n, r}$ that is vertical and transitive on fibers. The proof follow quickly from the next lemma.

**Lemma C.1.** *Suppose* $(U, P, V), (\widetilde{U}, \widetilde{P}, \widetilde{V}) \in \widetilde{M}_{m \times n, r}$ *are such that* $A := UPV^\top$. *The following hold:*

1. *A vector* $v \in \mathbb{R}^r$ *is an eigenvector of* $P$ *if and only if there exists a left (resp.right) eigenvectors* $x$ *of* $A$ *corresponding to a non-zero singular value of* $A$ *such that* $v = U^\top x$ *(resp. $v = V^\top x$). Moreover,* $UU^\top y = y$ *and* $VV^\top x = x$.

2. *Consequently, the non-zero singular values of* $A$ *are exactly the squared eigenvalues of* $P$, *and each non-zero singular value of* $A$ *has the same multiplicty as its corresponding eigenvalue of* $P$.

*Proof of Lemma C.1.* The second item follows immediately from the first one, so we only prove the first. First, suppose that $v \in \mathbb{R}^r$ is an eigenvector of $P$ with associated eigenvalue $\lambda > 0$. As $U^\top$ and $V^\top$ have full row rank, their columns span $\mathbb{R}^r$. Thus, there exist $x$ and $y$ such that $v = U^\top y$ and $v = V^\top x$. We have that

$$A^\top A x = V P^2 V^\top x = P^2 v = \lambda^2 v$$
$$AA^\top y = U P^2 U^\top y = P^2 v = \lambda^2 v,$$

so $y$ and $x$ are left and right eigenvectors associated to the non-zero singular value $\lambda^2$.

Next, suppose that $x$ is a left eigenvector associated to a non-zero singular value $\lambda^2 > 0$. We claim that $v = U^\top x$ is a eigenvector of $P$. To see this, we compute

$$\lambda^2 \|v\| = \lambda^2 \|x\| = x^\top AA^\top x = x^\top UP^2 U^\top x = v^\top P^2 v,$$

we have applied $\|v\| = \|x\|$, because $U \in \text{St}(m, r)$, in the first equality. Thus $V^\top x$ is an eigenvector of $P^2$ with eigenvalue $\lambda^2$. The eigenvectors of $P^2$ are coincide with those of $P$, because $P$ is positive semidefinite, so $V^\top x$ is an eigenvector of $P$ with eigenvalue $\lambda$. Replacing $x$ with $y$, $V$ with $U$, and $AA^\top$ with $A^\top xA$ in the above computation proves the claim for right eigenvectors.

Finally, with the equivalence established, we prove $UU^\top y = y$ and $VV^\top x = x$. We compute

$$\lambda^2 y = UP^2 U^\top y = \lambda^2 UU^\top y$$
$$\lambda^2 x = VP^2 V^\top x = \lambda^2 VV^\top x.$$

Dividing by $\lambda^2$ clinches the equalities.

$\square$

**Lemma C.2.** *The following hold for the fibers of $\pi$:*

1. *If $A \in \mathbb{R}_r^{m \times n}$ has the singular value decomposition $UDV^\top$ then $\pi^{-1}(A) = \{(UW, W^\top DW, VW) : W \in O(r)\}$.*

2. *$O(r)$, a Lie group, smooth and vertically acts on $\widetilde{M}_{m \times n, r}$ as $(U, P, V) \cdot W := (UW, W^\top PW, VW)$.*

3. *$O(r)$ acts transitively on fibers, and each fiber is properly embedded submanifold diffeomorphic to $O(r)$.*

*Proof of Lemma C.2.* We shall prove each item separately.

1. The inclusion $\{(UW, W^\top DW, VW) : W \in O(r)\} \subseteq \pi^{-1}(A)$ is an immediate consequence of matrix multiplication, so we only prove the reverse inclusion. Choose $(\widetilde{U}, P, \widetilde{V}) \in \text{St}(m, r) \times \mathbb{S}_{++}^r \times \text{St}(n, r)$ such that $\widetilde{U}P\widetilde{V}^\top = A$. Let $W = U^\top \widetilde{U}$. By construction, the columns of $U$ and $V$ are respectively the left and right eigenvectors of $A$ corresponding to non-zero singular values by construction. It thus follows from Lemma C.1, that the columns $\widetilde{U}^\top U$ and $\widetilde{V}^\top V$ are respectively the left and right eigenvectors of $P$, and, for each $i$, the $i$-th columns of $\widetilde{U}^\top U$ and $\widetilde{V}^\top V$ equal the same eigenvector of $P$. This latter fact is restated as $W = U^\top \widetilde{U} = V^\top \widetilde{V}$. Additionally, the first item of this last lemma implies $\widetilde{U}W^\top = \widetilde{U}\widetilde{U}^\top U = U$ and $\widetilde{V}W^\top = \widetilde{V}\widetilde{V}^\top V = V$, from which we see
$$I_r = U^\top U = U^\top \widetilde{U}\widetilde{U}^\top U = (\widetilde{U}^\top U)^\top \widetilde{U}^\top U = W^\top W,$$
so $W \in O(r)$. We then have
$$P = \widetilde{U}^\top A\widetilde{V} = (\widetilde{U}^\top U)D(V^\top \widetilde{V}) = W^\top DW.$$

2. The action of $O(r)$ on $\mathbb{R}^{m \times r} \times \mathbb{R}^{r \times r} \times \mathbb{R}^{n \times r}$ defined as $(U, P, V) \cdot W := (UW, W^\top PW, VW)$ is readily seen to be smooth because it is defined by matrix multiplications and transpositions. Being a product of embedded submanifolds of $\mathbb{R}^{m \times r} \times \mathbb{R}^{r \times r} \times \mathbb{R}^{n \times r}$, $\widetilde{M}_{m \times n, r}$ is an embedded submanifold. Thus, $O(r)$ restricts to a smooth action on this set. By the first item, this action is vertical.

3. By the first item, $O(r)$ is transitive on fibers. For a fixed $(U, P, V) \in \widetilde{M}_{m \times n, r}$, the map $W \in O(r) \mapsto (UW, W^\top PW, VW)$ is bijective and smooth, so it is an immersion by (Lee, 2012, Proposition 7.26). AS $O(r)$ is compact, this map is smooth embedding by (Lee, 2012, Proposition 4.22), and by (Lee, 2012, Corollary 5.6) it is proper embedding. $\square$

Now, we are fully equipped to prove our main lemma regarding the Riemannian quotient structure for $\mathbb{R}^{m \times n, r}$.

**Lemma 4.1** (Riemannian Quotient Structure for $\mathbb{R}_r^{m \times n}$). *The map $\pi$ is a Riemannian quotient map when $\mathbb{R}_r^{m \times n}$ is endowed with its embedded smooth structure and suitable Riemannian metric. Additionally, the vertical space at $(U, P, V) \in \widetilde{M}_{m \times n, r}$ is given by*

$$\mathcal{V}_{(U,P,V)} = \{(UA, PA - AP, VA) : A \in \mathrm{Skew}(r)\}.$$

*Proof of Lemma 4.1.* We follow a two step process:

1. The map $\pi$ is a smooth submersion. This requires only that we show the vertical space at $(U, P, V) \in \widetilde{M}_{m \times n, r}$ is given by

$$\mathcal{V}_{(U,P,V)} = \{(UA, PA - AP, VA) : A \in \mathrm{Skew}(r)\},$$

because the vertical spaces the same dimension across $\widetilde{M}_{m \times n, r}$. The map $\pi$ is surjective so the Global Rank Theorem (Lee, 2012, Theorem 4.14) implies $\pi$ is a smooth submersion.

2. The group $O(r)$ acts isometrically on $\widetilde{M}_{m \times n, r}$ given the metric (7c). This fact, in conjunction with Lemma C.2 and (Lee, 2012, Theorem 2.28), implies that $\mathbb{R}_r^{m \times}$ can be endowed with a metric making $\pi$ is a Riemannian quotient map.

1. Fix $(U, P, V) \in \widetilde{M}_{m \times n, r}$. Let

$$\tilde{\mathcal{V}}_{U,P,V} = \{(UA, PA - AP, VA) : A \in \mathrm{Skew}(r)\}.$$

We need to show that $\tilde{\mathcal{V}}_{U,P,V} = \mathcal{V}_{U,P,V}$. The inclusion, $\tilde{\mathcal{V}}_{U,P,V} \subseteq \mathcal{V}_{U,P,V}$, is relatively straightforward. The differential of the diffeomorphism $W \in O(r) \mapsto (UW, W^\top PW, VW)$ at $I_r$ is

$$H \mapsto (UH, PH - HP, VH)$$

for $H \in T_{I_r} O(r) = \mathrm{Skew}(r)$. As the map is a diffeomorphism, the differential is an isomorphism onto $T_{(U,P,V)} \pi^{-1}(UPV^\top) = \mathrm{Ker}(d\pi_{(U,P,V)})$, and the inclusion follows.

We move onto the next inclusion. Suppose that $(H_1, D, H_2) \in T_{(U,P,V)} \widetilde{M}_{m \times n, r}$ such that $d\pi_{(U,P,V)}(H_1, D, H_2) = 0$. Consider a curve $t \mapsto (U + tH_1, P + tD, V + tH_2)$, and the corresponding curve

$$t \mapsto \pi(U + tH_1, P + tD, V + H_2) = (U + tH_1)(P + tD)(U + tH_2)^\top.$$

We are permitted to consider these curves in computing $\ker(d\pi_{(U,P,V)})$ because $\widetilde{M}_{m \times n, r}$ and $\mathbb{R}_r^{m \times n}$ are properly embedded submanifolds of their ambient Euclidean spaces. These proper embeddings allow us to consider $\pi$'s extension to the entire ambient space of $\widetilde{M}_{m \times n, r}$, which has codomain $\mathbb{R}^{m \times n}$, when computing $\ker(d\pi_{(U,P,V)})$. Taking derivatives at $t = 0$, we see $d\pi_{(U,P,V)}(U + tH_1, P + tD, V + H_2) = \frac{d}{dt} \pi(U + tH_1, P + tD, V + H_2)|_{t=0}$, which produces

$$d\pi_{(U,P,V)}(H_1, D, H_2) = H_1 PV^\top + UDV^\top + UPH_2^\top.$$

There exist $A_1, A_2 \in \mathrm{Skew}(r)$, $B_1 \in \mathbb{R}^{(n-r) \times r}$, and $B_2 \in \mathbb{R}^{(m-r) \times r}$ such that $H_1 = UA_1 + U^\perp B_1$ and $H_2 = VA_2 + V^\perp B_2$. Plugging these into the above equation, we find the condition

$$0 = d\pi_{(U,P,V)}(H_1, D, H_2) = (UA_1 + U^\perp B_1)PV^\top + UDV^\top + UP(VA_2 + V^\perp B_2)^\top$$
$$= U(A_1 P + PA_2^\top + D)V^\top + U^\perp B_1 PV^\top + UPB_2^\top(V^\perp)^\top$$

Multiplying on the left and the right by $U^\top$ and $V^\perp$ respectively, then by $(U^\perp)^\top$ and $V$ respectively, we obtain $B_1 = 0$ and $B_2 = 0$, so the above equation becomes

$$0 = U(A_1 P + PA_2^\top + D)V^\top.$$

Multiplying on the left and the right by $U^\top$ and $V$ respectively, we obtain $D = -(A_1 P + PA_2^\top)$. This implies $A_1 P - PA_2$ is symmetric, i.e.

$$A_1 P - PA_2 = (A_1 P - PA_2)^\top = -PA_1 + A_2 P$$

which can be rearranged to

$$(A_1 - A_2)P + P(A_1 - A_2) = 0.$$

Multiplying this by any eigenvector $q$ of $P$, with corresponding eigenvalue $\lambda$, we see

$$\begin{aligned} 0 &= [(A_1 - A_2)P + P(A_1 - A_2)]q \\ &= \lambda(A_1 - A_2)q + P(A_1 - A_2)q \\ &= [\lambda I_r + P](A_1 - A_2)q. \end{aligned}$$

The matrix, $\lambda I_r + P$, is positive definite so it must hold that $= 0(A_1 - A_2)q$. As $q$ is arbitrary eigenvector of $P$, and the eigenvectors of $P$ form a basis for $\mathbb{R}^r$, we conclude $A_1 - A_2 = 0$ as desired. We thus, conclude that $(H_1, D, H_2)$ has the requisite form to belong to $\tilde{\mathcal{V}}_{(U,P,V)}$.

2. We need only show that $O(r)$ acts isometrically on $M$ given the metric (7c). The metric is product metric so it suffices to show each of the constituent metrics are invariant under $O(r)$'s action on each component. This follows from the relatively simple computations below.

- $\mathrm{St}(n,r)$: Choose $U \in \mathrm{St}(n,r)$ and $W \in O(r)$. Furthermore, pick $UA_1 + U^\perp B_1, UA_2 + U^\perp B_2 \in T_U \mathrm{St}(n,r)$. We compute

$$\begin{aligned} &\left\langle UA_1 + U^\perp B_1, UA_2 + U^\perp B_2 \right\rangle_U^{\mathrm{St}(n,r)} \\ &= \frac{1}{2}\langle A_1, A_2 \rangle + \langle B_1, B_2 \rangle \\ &= \frac{1}{2}\langle WA_1, WA_2 \rangle + \langle WB_1, WB_2 \rangle \\ &= \left\langle (UW)A_1 + (U^\perp W)B_1, (UW)A_2 + (U^\perp W)B_2 \right\rangle_{UW}^{\mathrm{St}(n,r)} \end{aligned}$$

- $\mathbb{S}_{++}^r$: Choose $P \in \mathbb{S}_{++}^r$ and $W \in O(r)$. Furthermore, pick $D_1, D_2 \in T_P \mathbb{S}_{++}^r = \mathbb{S}^r$. We compute

$$\begin{aligned} \langle D_1, D_2 \rangle_P^{\mathbb{S}_{++}^r} &= \langle P^{-1/2}D_1 P^{-1/2}, P^{-1/2}D_2 P^{-1/2} \rangle \\ &= \Big\langle (W^\top P^{-1/2}W)(W^\top D_1 W)(W^\top P^{-1/2}W), \\ &\qquad (W^\top P^{-1/2}W)(W^\top D_2 W)(W^\top P^{-1/2}W) \Big\rangle \\ &= \langle W^\top D_1 W, W^\top D_2 W \rangle_{W^\top PW}^{\mathbb{S}_{++}^r} \end{aligned}$$

$\square$

## C.2    Horizontal Subspace Decomposition

This subsection focuses on the proofs of Lemma 4.2, Proposition 4.3, and Proposition 4.4.

**Lemma 4.2** (Horizontal Space for $\mathbb{R}_r^{m \times n}$). *For any $(U, P, V) \in \widetilde{M}_{m \times n, r}$, we have*

$$\mathcal{H}_{(U,P,V)} = \Big\{ Z(A, D, B_1, B_2) : B_1 \in \mathbb{R}^{(m-r) \times r}, B_2 \in \mathbb{R}^{(n-r) \times r}, D \in \mathbb{S}^r, A \in \mathrm{Skew}(r) \Big\}$$

*where*

$$\begin{aligned} &Z(A, D, B_1, B_2) \\ &:= \left( U(P^{-1}D - DP^{-1} + A) + U^\perp B_1, \frac{1}{2}D, V(P^{-1}D - DP^{-1} - A) + V^\perp B_2 \right). \end{aligned}$$

*Proof of Lemma 4.2.* Select an arbitrary tangent vector

$$(U\bar{A}_1 + U^\perp \bar{B}_1, \bar{D}, V\bar{A}_2 + V^\perp \bar{B}_2) \in T_{(U,P,V)} \widetilde{M}_{m \times n, r}.$$

We shall find conditions on $\bar{A}_1, \bar{B}_1, \bar{A}_2, \bar{B}_2, \bar{D}$ so that

$$\left\langle (U\bar{A}_1 + U^\perp \bar{B}_1, \bar{D}, V\bar{A}_2 + V^\perp \bar{B}_2), (UA, PA - AP, VA) \right\rangle_{U,P,V} = 0$$

for all $A \in \mathrm{Skew}(r)$. The inner product can be written as

$$\left\langle (U\bar{A}_1 + U^\perp \bar{B}_1, \bar{D}, V\bar{A}_2 + V^\perp \bar{B}_2), (UA, PA - AP, VA) \right\rangle_{U,P,V}$$

$$= \frac{1}{2}\langle \bar{A}_1, A \rangle + \langle P^{-1}\bar{D}P^{-1}, PA - AP \rangle + \frac{1}{2}\langle \bar{A}_2, A \rangle$$

$$= \frac{1}{2}\langle \bar{A}_1 + \bar{A}_2, A \rangle + \langle \bar{D}P^{-1}, A \rangle - \langle P^{-1}\bar{D}, A \rangle$$

$$= \left\langle \frac{1}{2}(\bar{A}_1 + \bar{A}_2) - \left(P^{-1}\bar{D} - \bar{D}P^{-1}\right), A \right\rangle.$$

Therefore the condition we need is

$$\frac{1}{2}(\bar{A}_1 + \bar{A}_2) - \left(P^{-1}\bar{D} - \bar{D}P^{-1}\right) \in \mathbb{S}^r.$$

But both matrices in the sum above are skew-symmetric, which implies this matrix is orthogonal to $\mathbb{S}^r$, hence we have $\bar{A}_1 + \bar{A}_2 = 2(P^{-1}\bar{D} - \bar{D}P^{-1})$. This gives the horizontal space. $\qquad\square$

**Proposition 4.3** (Horizontal Decomposition for $\mathbb{R}_r^{m \times n}$). *Let $(U, P, V) \in \widetilde{M}_{m \times n, r}$ and $P = Q\Lambda Q^\top$ be a spectral decomposition of $P$. The collection of horizontal subspaces*

$$\begin{aligned}
S_{1,k}(U,P,V) &:= \left\{ (uq_k^\top, 0, 0) : u \in \ker(U^\top) \right\}, & k \in [r] \\
S_{2,k}(U,P,V) &:= \left\{ (0, 0, vq_k^\top) : v \in \ker(V^\top) \right\}, & k \in [r] \\
S_{\mathbb{S},ij}(U,P,V) &:= \left\{ \tau Z(D_{ij}^P, 0, 0, 0) : \tau \in \mathbb{R} \right\}, & 1 \le i, j \le r \\
S_{\mathrm{Skew},k\ell}(U,P,V) &:= \left\{ \tau \left(UH_{k\ell}, 0, -VH_{k\ell}\right) : \tau \in \mathbb{R} \right\}, & 1 \le k < \ell \le r
\end{aligned}$$

*where $Z$ is as in Lemma 4.2, forms an horizontal subspace decomposition of $\mathcal{H}_{(U,P,V)}$ under the metric* (7c). *Moreover, there are $m = \frac{r(2r+3)}{2}$ different subspaces in total, so the subspace selection rule that chooses uniformly from this collection satisfies Assumption 3.2 with $C = \frac{1}{\sqrt{m}} = \sqrt{\frac{2}{r(2r+3)}}$.*

*Proof of Proposition 4.3.* It is clear that

- $S_{1,k}(U,P,V)$ and $S_{2,k'}(U,P,V)$ are orthogonal to one another, and are orthogonal to $S_{\mathbb{S},ij}(U,P,V)$ and $S_{\mathrm{Skew},i'j'}(U,P,V)$

- the collections $\{S_{1,k}(U,P,V) : k \in [r]\}$ and $\{S_{2,k}(U,P,V) : k \in [r]\}$ are individually, mutually orthogonal.

Consequently, we must prove that

- $S_{\mathbb{S},ij}(U,P,V)$ and $S_{\mathbb{S},k\ell}(U,P,V)$ are orthogonal when $(i,j) \neq (k,\ell)$

- $S_{\mathrm{Skew},ij}(U,P,V)$ and $S_{\mathrm{Skew},k\ell}(U,P)$ are orthogonal when $(i,j) \neq (k,\ell)$

- $S_{\mathbb{S},ij}(U,P,V)$ and $S_{\mathrm{Skew},k\ell}(U,P,V)$ are orthogonal

First, we will prove $S_{\mathbb{S},ij}(U,P,V)$ and $S_{\mathbb{S},k\ell}(U,P,V)$ are orthogonal when $(i,j) \neq (k,\ell)$. To see this, observe that

$$\left\langle \left(U(P^{-1}D_{ij}^P - D_{ij}^P P^{-1}), \frac{1}{2}D_{ij}^P, V(P^{-1}D_{ij}^P - D_{ij}^P P^{-1})\right), \left(U(P^{-1}D_{kl}^P - D_{kl}^P P^{-1}), \frac{1}{2}D_{kl}^P\right) \right\rangle_{U,P,V}$$

$$= \mathrm{Tr}\left(\left(P^{-1}D_{ij}^P - D_{ij}^P P^{-1}\right)^\top \left(P^{-1}D_{kl}^P - D_{kl}^P P^{-1}\right)\right)$$

$$+ \frac{1}{4}\mathrm{Tr}\left(\left(P^{-1/2}D_{ij}^P P^{-1/2}\right)^\top \left(P^{-1/2}D_{kl}^P P^{-1/2}\right)\right).$$

Denote $S_{ij} = q_i q_j^\top + q_j q_i^\top$, and analogously for $S_{kl}$. For the second term, notice that by definition, we have $P^{-1/2} D_{ij}^P P^{-1/2} = S_{ij}$, and analogously for $P^{-1/2} D_{kl}^P P^{-1/2}$. Therefore

$$\mathrm{Tr}\left(\left(P^{-1/2} D_{ij}^P P^{-1/2}\right)^\top \left(P^{-1/2} D_{kl}^P P^{-1/2}\right)\right) = \mathrm{Tr}(S_{ij} S_{kl}).$$

This is $0$ unless $(i,j) = (k,l)$ or $(i,j) = (l,k)$.

For the first term, notice that $P^{-1} D_{ij}^P - D_{ij}^P P^{-1} = (\lambda_i \lambda_i)^{1/2}(P^{-1} S_{ij} - S_{ij} P^{-1}) = (\lambda_i \lambda_j)^{1/2}(\lambda_i^{-1} q_i q_j^\top + \lambda_j^{-1} q_j q_i^\top - \lambda_j^{-1} q_i q_j^\top - \lambda_i^{-1} q_j q_i^\top) = (\lambda_i \lambda_j)^{1/2}(\lambda_i^{-1} - \lambda_j^{-1})(q_i q_j^\top - q_j q_i^\top)$. Denote $H_{ij} = q_i q_j^\top - q_j q_i^\top$ and analogously for $H_{kl}$. Therefore

$$\left(P^{-1} D_{ij}^P - D_{ij}^P P^{-1}\right)^\top \left(P^{-1} D_{kl}^P - D_{kl}^P P^{-1}\right)$$
$$= (\lambda_i \lambda_j)^{1/2}(\lambda_i^{-1} - \lambda_j^{-1})(\lambda_k \lambda_l)^{1/2}(\lambda_k^{-1} - \lambda_l^{-1}) H_{ij}^\top H_{kl}.$$

Therefore

$$\mathrm{Tr}\left(\left(P^{-1} D_{ij}^P - D_{ij}^P P^{-1}\right)^\top \left(P^{-1} D_{kl}^P - D_{kl}^P P^{-1}\right)\right)$$
$$= (\lambda_i \lambda_j)^{1/2}(\lambda_i^{-1} - \lambda_j^{-1})(\lambda_k \lambda_l)^{1/2}(\lambda_k^{-1} - \lambda_l^{-1}) \mathrm{Tr}(H_{ij}^\top H_{kl}).$$

This is equal to $0$ unless $(i,j) = (k,l)$ or $(i,j) = (l,k)$.

Therefore, each term in the inner product is zero unless $(i,j) = (k,l)$ or $(i,j) = (l,k)$. However, since $i \le j$, $k \le l$, this only occurs if $i = k$, $j = l$. Therefore the subspaces $S_{ij}(U, P)$ are mutually orthogonal.

Next, we will prove that $S_{\mathrm{Skew},ij}(U, P, V)$ and $S_{\mathrm{Skew},k\ell}(U, P, V)$ are orthogonal when $(i,j) \ne (k,\ell)$. We compute

$$\langle (UH_{ij}, 0, -VH_{ij}), (UH_{kl}, 0, -VH_{kl}) \rangle = 2\,\mathrm{Tr}(H_{ij} H_{kl})$$

which again equals zero if $(i,j) \ne (k,l)$.

Finally, we will prove that $S_{\mathbb{S},ij}(U, P, V)$ and $S_{\mathrm{Skew},k\ell}(U, P, V)$ are orthogonal. We compute

$$\left\langle \left(U(P^{-1} D_{ij}^P - D_{ij}^P P^{-1}), \frac{1}{2} D_{ij}^P, V(P^{-1} D_{ij}^P - D_{ij}^P P^{-1})\right), (UH_{kl}, 0, -VH_{kl}) \right\rangle_{U,P,V}$$
$$= \frac{1}{2}\left[\mathrm{Tr}\left((P^{-1} D_{ij}^P - D_{ij}^P P^{-1})H_{kl}\right) - \mathrm{Tr}\left((P^{-1} D_{ij}^P - D_{ij}^P P^{-1})H_{kl}\right)\right] = 0,$$

which establishes the claim.

To see why Assumption 3.2 is satisfied, notice that the subspaces span $\mathcal{H}_{(U,P,V)}$, so

$$\mathcal{H}_{(U,P,V)} = \left(\bigoplus_{k \in [r]} S_{1,k}(U, P, V)\right) \oplus \left(\bigoplus_{1 \le i \le j \le r} S_{\mathrm{Skew},ij}(U, P, V)\right)$$
$$\oplus \left(\bigoplus_{1 \le i \le j \le r} S_{\mathbb{S},ij}(U, P, V)\right) \oplus \left(\bigoplus_{k \in [r]} S_{2,k}(U, P, V)\right)$$

and they are orthogonal. Therefore for any $v \in \mathcal{H}_{(U,P)}$, we have

$$
\begin{aligned}
\|v\|_{U,P} &= \Bigg( \sum_{k \in [r]} \left[ \left\| \mathrm{Proj}_{S_{1,k}(U,P,V)}(v) \right\|_{U,P,V}^2 + \left\| \mathrm{Proj}_{S_{2,k}(U,P,V)}(v) \right\|_{U,P,V}^2 \right] \\
&\qquad + \sum_{1 \le i \le j \le r} \left[ \left\| \mathrm{Proj}_{S_{\mathbb{S},ij}(U,P,V)}(v) \right\|_{U,P,V}^2 + \left\| \mathrm{Proj}_{S_{\mathrm{Skew},ij}(U,P,V)}(v) \right\|_{U,P,V}^2 \right] \Bigg)^{\frac{1}{2}} \\
&= \Bigg\{ \frac{r(r+3)}{r(r+3)} \Bigg[ \sum_{k \in [r]} \left( \left\| \mathrm{Proj}_{S_{1,k}(U,P,V)}(v) \right\|_{U,P,V}^2 + \left\| \mathrm{Proj}_{S_{2,k}(U,P,V)}(v) \right\|_{U,P,V}^2 \right) \\
&\qquad + \sum_{1 \le i \le j \le r} \left( \left\| \mathrm{Proj}_{S_{\mathbb{S},ij}(U,P,V)}(v) \right\|_{U,P,V}^2 + \left\| \mathrm{Proj}_{S_{\mathrm{Skew},ij}(U,P,V)}(v) \right\|_{U,P,V}^2 \right) \Bigg] \Bigg\}^{\frac{1}{2}} \\
&= \sqrt{[r(r+3)] \cdot \mathbb{E}\left[ \left\| \mathrm{Proj}_{S(U,P,V)}(v) \right\|_{U,P,V}^2 \,\Big|\, U, P, V \right]},
\end{aligned}
$$

thus Assumption 3.2 is satisfied with $C = \frac{1}{\sqrt{m}} = \sqrt{\frac{2}{r(2r+3)}}$. $\qquad\square$

We conclude with a proof of non-smoothness of the horizontal subspace decomposition.

**Proposition 4.4** (Non-smooth Randomized Subspace Selection for $\mathbb{R}_r^{m \times n}$). *If $r > 2$, then the subspace selection rule induced by the horizontal decomposition (8) is not smooth.*

*Proof of Proposition 4.4.* It suffices to show a single subspace from the decomposition is non-smooth. We consider $S_{\mathbb{S},12}(U, P, V)$ which is spanned by

$$
W(U, P, V) = \left( \left( \frac{1}{\lambda_1} - \frac{1}{\lambda_2} \right) U H_{12}, \frac{1}{2}(q_1 q_2^\top + q_2 q_1^\top), \left( \frac{1}{\lambda_1} - \frac{1}{\lambda_2} \right) V H_{12} \right).
$$

where $q_1$ and $q_2$ are eigenvectors with corresponding eigenvalues $\lambda_1$ and $\lambda_2$. In fact, as $S_{\mathbb{S},12}(U, P, V)$ has dimension equal to 1, any spanning vector of it must be a scalar multiple of $W(U, P, V)$. Set

$$
U_0 = \begin{bmatrix} I_r \\ 0 \end{bmatrix}, \quad P_0 = I_r, \quad V_0 = \begin{bmatrix} I_r \\ 0 \end{bmatrix},
$$

$q_1 = e_1$, and $q_2 = e_2$. Then

$$
W(U_0, P_0, V_0) = \left( 0, \frac{1}{2}(e_1 e_2^\top + e_2 e_1^\top), 0 \right).
$$

Observe the eigenvectors for $P + \epsilon(e_1 e_2^\top + e_2 e_1^\top)$ are

$$
\frac{1}{\sqrt{2}}(e_1 + e_2), \frac{1}{\sqrt{2}}(e_1 - e_2), e_3, \ldots, e_r
$$

and the eigenvectors for $P + \epsilon(e_1 e_3^\top + e_3 e_1^\top)$ are

$$
\frac{1}{\sqrt{2}}(e_1 + e_3), \frac{1}{\sqrt{2}}(e_1 - e_3), e_2, e_4, \ldots, e_r
$$

Consequently, there is no selection of eigenvectors of $P$ that ensures $q_1$ and $q_2$ are continuous, $\lim_{P \to I_r} q_1(P) = e_1$, and $\lim_{P \to I_r} q_2(P) = e_2$. Thus, $W$ is discontinuous at $(U_0, P_0, V_0)$. $\qquad\square$

## C.3 Efficient Update Formulas: Exponential Maps & Gradient Projections

In the main text, we presented streamlined statements for Theorem 4.5 and Proposition 4.6. In this section, we dive into the efficient expressions for the exponential map and the gradient projections. We provide complete statements and proofs of these results.

**Theorem 4.5** (Restated). *Fix $\tau \in \mathbb{R}$. Let $(U, P, V) \in \widetilde{M}_{m \times n, r}$ with $P = Q\Lambda Q^\top$ being a spectral decomposition for $P$, where $Q = [q_1 \quad \cdots \quad q_r] \in O(r), \Lambda = \mathrm{Diag}(\lambda_1, \ldots, \lambda_r) \succ 0$. Let $W \in T_{(U,P,V)}\widetilde{M}_{m \times n, r}$, and for $\tau \in \mathbb{R}$, we define $U_+(\tau), P_+(\tau), V_+(\tau), Q_+(\tau), \Lambda_+(\tau)$ as follows:*

$$(U_+(\tau), P_+(\tau), V_+(\tau)) = \widetilde{\mathrm{Exp}}_{(U,P,V)}(\tau W), \quad P_+(\tau) = Q_+(\tau)\Lambda_+(\tau)Q_+^\top(\tau).$$

*Then we have the following update rules:*

1. *Fix any $k \in [r]$ and $u \in \ker(U^\top)$. If $W = (uq_k^\top, 0, 0) \in S_{1,k}(U, P, V)$, then*

$$U_+(\tau) = U(I - q_k q_k^\top) + \left( \cos(\tau\|u\|_2)Uq_k + \sin(\tau\|u\|_2)\frac{u}{\|u\|_2} \right) q_k^\top,$$

   *while $(P_+(\tau), V_+(\tau), Q_+(\tau), \Lambda_+(\tau)) = (P, V, Q, \Lambda)$. This means that $U_+(\tau)Q_+(\tau)$ is obtained by replacing the $k$th column of $UQ$, which is $Uq_k$, with a linear combination $\cos(\tau\|u\|_2)Uq_k + \sin(\tau\|u\|_2)\frac{u}{\|u\|_2}$.*

2. *Fix any $k \in [r]$ and $v \in \ker(V^\top)$. If $W = (0, 0, vq_k^\top) \in S_{2,k}(U, P, V)$, then*

$$V_+(\tau) = V(I - q_k q_k^\top) + \left( \cos(\tau\|v\|_2)Vq_k + \sin(\tau\|v\|_2)\frac{v}{\|v\|_2} \right) q_k^\top,$$

   *while $(U_+(\tau), P_+(\tau), Q_+(\tau), \Lambda_+(\tau)) = (U, P, Q, \Lambda)$. This means that $V_+(\tau)Q_+(\tau)$ is obtained by replacing the $k$th column of $VQ$, which is $Vq_k$, with a linear combination $\cos(\tau\|v\|_2)Vq_k + \sin(\tau\|v\|_2)\frac{v}{\|v\|_2}$.*

3. *Fix any $1 \le i < j \le r$. Suppose $W = (U(P^{-1}D_{ij}^P - D_{ij}^P P^{-1}), \frac{1}{2}D_{ij}^P, U(P^{-1}D_{ij}^P - D_{ij}^P P^{-1})) \in S_{\mathbb{S},ij}(U, P, V)$. Let $\alpha_{ij}(\tau) := \tau(\lambda_i \lambda_j)^{1/2}(\lambda_i^{-1} - \lambda_j^{-1})$. Then*

$$\begin{aligned}
U_+(\tau) &= U(I - q_i q_i^\top - q_j q_j^\top) + (\cos(\alpha_{ij}(\tau))Uq_i - \sin(\alpha_{ij}(\tau))Uq_j)q_i^\top \\
&\quad + (\sin(\alpha_{ij}(\tau))Uq_i + \cos(\alpha_{ij}(\tau))Uq_j)q_j^\top \\
P_+(\tau) &= P - (\lambda_i q_i q_i^\top + \lambda_j q_j q_j^\top) \\
&\quad + \cosh(\tau/2)(\lambda_i q_i q_i^\top + \lambda_j q_j q_j^\top) + \sinh(\tau/2)\sqrt{\lambda_i \lambda_j}(q_i q_j^\top + q_j q_i^\top) \\
V_+(\tau) &= V(I - q_i q_i^\top - q_j q_j^\top) + (\cos(\alpha_{ij}(\tau))Vq_i - \sin(\alpha_{ij}(\tau))Vq_j)q_i^\top \\
&\quad + (\sin(\alpha_{ij}(\tau))Vq_i + \cos(\alpha_{ij}(\tau))Vq_j)q_j^\top.
\end{aligned}$$

   *Define*

$$M_{ij}(\tau) := \begin{bmatrix} \cosh(\tau/2)\lambda_i & \sinh(\tau/2)\sqrt{\lambda_i \lambda_j} \\ \sinh(\tau/2)\sqrt{\lambda_i \lambda_j} & \cosh(\tau/2)\lambda_j \end{bmatrix}, \quad G(\tau) := \begin{bmatrix} \cos(\alpha_{ij}(\tau)) & \sin(\alpha_{ij}(\tau)) \\ -\sin(\alpha_{ij}(\tau)) & \cos(\alpha_{ij}(\tau)) \end{bmatrix}.$$

   *Let $e_1(\tau), e_2(\tau) \in \mathbb{R}^2$ be the two eigenvectors of $M_{ij}(\tau)$ with corresponding eigenvalues $\gamma_1(\tau), \gamma_2(\tau)$. Define $Q_{ij} := [q_i \quad q_j]$. Only the $i$ and $j$ columns of $U_+(\tau)Q_+(\tau)$ and $V_+(\tau)Q_+(\tau)$ are changed; all others remain the same as those of $UQ$ and $VQ$. These columns are updated as follows:*

$$\begin{aligned}
Uq_i &\to UQ_{ij}G(\tau)e_1(\tau), \quad Uq_j \to UQ_{ij}G(\tau)e_2(\tau) \\
Vq_i &\to VQ_{ij}G(\tau)e_1(\tau), \quad Vq_j \to VQ_{ij}G(\tau)e_2(\tau).
\end{aligned}$$

   *(Here, $UQ_{ij}$ is an $m \times 2$ matrix consisting of the $i$ and $j$ columns of $UQ$, and similarly for $VQ_{ij}$.) Furthermore, in $\Lambda_+(\tau)$, the entry $\lambda_i$ is replaced with $\gamma_1(\tau)$, while $\lambda_j$ is replaced with $\gamma_2(\tau)$, and all other entries remain the same.*

4. *Fix any $i \in [r]$. If $W = \left( U(P^{-1}D_{ii}^P - D_{ii}^P P^{-1}), \frac{1}{2}D_{ii}^P, V(P^{-1}D_{ii}^P - D_{ii}^P P^{-1}) \right) \in S_{\mathbb{S},ii}(U, P, V)$, then $W = (0, D_{ii}^P/2, 0)$. Thus*

$$P_+(\tau) = P - \lambda_i q_i q_i^\top + \lambda_i \exp(\tau)q_i q_i^\top.$$

   *while $(U_+(\tau), V_+(\tau), Q_+(\tau)) = (U, V, Q)$. This means that $\lambda_i$ is replaced with $\lambda_i \exp(\tau)$ in $\Lambda_+$, while all other entries are the same.*

5. *Fix any $1 \leq k < \ell \leq r$. If $W = (U H_{k\ell}, 0, -V H_{k\ell}) \in S_{\text{Skew},k\ell}$, then we have*

$$U_+(\tau) = U(I - q_k q_k^\top - q_\ell q_\ell^\top)$$
$$+ (\cos(\tau) U q_k - \sin(\tau) U q_\ell) \, q_k^\top + (\sin(\tau) U q_k + \cos(\tau) U q_\ell) \, q_\ell^\top$$
$$V_+(\tau) = U(I - q_k q_k^\top - q_\ell q_\ell^\top)$$
$$+ (\cos(\tau) V q_k + \sin(\tau) V q_\ell) \, q_k^\top + (- \sin(\tau) V q_k + \cos(\tau) V q_\ell) \, q_\ell^\top,$$

*while $P_+(\tau) = P$. Thus, in $U_+(\tau) Q_+(\tau)$, $U q_k$ is replaced with $\cos(\tau) U q_k - \sin(\tau) U q_\ell$ and $U q_\ell$ is replaced with $\sin(\tau) U q_k + \cos(\tau) U q_\ell$. An analogous update holds for $V_+(\tau) Q_+(\tau)$.*

*Proof of Theorem 4.5.* We proceed item by item.

1 and 2. This result follows from (Gutman & Ho-Nguyen, 2023, Lemma 5).

3. Note from the proof of Proposition 5.3, we have $P^{-1/2} D_{ij}^P P^{-1/2} = S_{ij}$ and $P^{-1} D_{ij}^P - D_{ij}^P P^{-1} = (\lambda_i \lambda_j)^{1/2} (\lambda_i^{-1} - \lambda_j^{-1}) H_{ij}$, where $S_{ij} = q_i q_j^\top + q_j q_i^\top$ and $H_{ij} = q_i q_j^\top - q_j q_i^\top$. Then by (11) we have

$$U_+(\tau) = U \operatorname{Expm}\left(\tau (\lambda_i \lambda_j)^{1/2} (\lambda_i^{-1} - \lambda_j^{-1}) H_{ij}\right)$$
$$P_+(\tau) = P^{1/2} \operatorname{Expm}((\tau/2) S_{ij}) P^{1/2},$$
$$V_+(\tau) = V \operatorname{Expm}\left(\tau (\lambda_i \lambda_j)^{1/2} (\lambda_i^{-1} - \lambda_j^{-1}) H_{ij}\right).$$

It is straightforward to check that $\operatorname{Expm}(\alpha H_{ij})$ is a Givens rotation in the subspace spanned by $q_i$ and $q_j$, i.e.,

$$\operatorname{Expm}(\alpha H_{ij}) = I - (q_i q_i^\top + q_j q_j^\top) + \cos(\alpha)(q_i q_i^\top + q_j q_j^\top) + \sin(\alpha)(q_i q_j^\top - q_j q_i^\top).$$

Therefore

$$U \operatorname{Expm}\left(\tau (\lambda_i \lambda_j)^{1/2} (\lambda_i^{-1} - \lambda_j^{-1}) H_{ij}\right)$$
$$= U(I - q_i q_i^\top - q_j q_j^\top) + (\cos(\alpha_{ij}) U q_i - \sin(\alpha_{ij}) U q_j) q_i^\top + (\sin(\alpha_{ij}) U q_i + \cos(\alpha_{ij}) U q_j) q_j^\top,$$
$$V \operatorname{Expm}\left(\tau (\lambda_i \lambda_j)^{1/2} (\lambda_i^{-1} - \lambda_j^{-1}) H_{ij}\right)$$
$$= V(I - q_i q_i^\top - q_j q_j^\top) + (\cos(\alpha_{ij}) V q_i - \sin(\alpha_{ij}) V q_j) q_i^\top + (\sin(\alpha_{ij}) V q_i + \cos(\alpha_{ij}) V q_j) q_j^\top,$$

where $\alpha_{ij} = \tau (\lambda_i \lambda_j)^{1/2} (\lambda_i^{-1} - \lambda_j^{-1})$.

We also have

$$\operatorname{Expm}(\alpha S_{ij}) = I - (q_i q_i^\top + q_j q_j^\top) + \cosh(\alpha)(q_i q_i^\top + q_j q_j^\top) + \sinh(\alpha)(q_i q_j^\top + q_j q_i^\top),$$

hence, with $\alpha = \tau/2$, we have

$$P^{1/2} \operatorname{Expm}(P^{-1/2}(\tau/2) D_{ij}^P P^{-1/2}) P^{1/2}$$
$$= P^{1/2} \operatorname{Expm}(\alpha S_{ij}) P^{1/2}$$
$$= P - (\lambda_i q_i q_i^\top + \lambda_j q_j q_j^\top) + \cosh(\alpha)(\lambda_i q_i q_i^\top + \lambda_j q_j q_j^\top) + \sinh(\alpha)\sqrt{\lambda_i \lambda_j}(q_i q_j^\top + q_j q_i^\top).$$

The updates can be written as

$$U_+(\tau) = U(I - Q_{ij} Q_{ij}^\top) + U Q_{ij} G(\tau) Q_{ij}^\top$$
$$P_+(\tau) = P(I - Q_{ij} Q_{ij}^\top) + Q_{ij} M_{ij}(\tau) Q_{ij}^\top$$
$$V_+(\tau) = V(I - Q_{ij} Q_{ij}^\top) + V Q_{ij} G(\tau) Q_{ij}^\top.$$

It is easy to check that if $k \neq i, j$, then $P_+(\tau) q_k = \lambda_k q_k$, hence $q_k$ is still an eigenvector of $P_+(\tau)$ with eigenvalue $\lambda_i$, so these components do not change in $Q_+(\tau), \Lambda_+(\tau)$. Furthermore,

if $M_{ij}(\tau)e_1(\tau) = \gamma_1(\tau)e_1(\tau)$, then we have $Q_{ij}M_{ij}(\tau)Q_{ij}^\top(Q_{ij}e_1(\tau)) = \gamma_1(\tau)Q_{ij}e_1(\tau)$, and similarly for $e_2(\tau)$. Therefore $Q_{ij}e_1(\tau), Q_{ij}e_2(\tau)$ are the new eigenvectors of $P_+(\tau)$, with eigenvalues $\gamma_1(\tau), \gamma_2(\tau)$. Correspondingly, the new columns of $U_+(\tau)Q_+(\tau)$ are $U_+(\tau)Q_{ij}e_1(\tau) = UQ_{ij}G(\tau)e_1(\tau)$ and $U_+(\tau)Q_{ij}e_2(\tau) = UQ_{ij}G(\tau)e_2(\tau)$.

4. Note that $D_{ij}^P = \lambda_i q_i q_i^\top$ and $\mathrm{Expm}\left(\tau q_i q_i^\top\right) = I - q_i q_i^\top + \exp(\tau)q_i q_i^\top$, by (11) we have

$$P_+(\tau) = P^{1/2}\,\mathrm{Expm}\left(\tau q_i q_i^\top\right)P^{1/2} = P - \lambda_i q_i q_i^\top + \lambda_i \exp\left(\tau\right)q_i q_i^\top.$$

All other components remain the same. Note that the eigenvectors of $P_+(\tau)$ are exactly the same as $P$. Furthermore, all eigenvalues are the same, except that $P_+(\tau)q_i = \lambda_i \exp\left(\tau\right)q_i$.

5. Observe that by (11) we have

$$U_+(\tau) = U\,\mathrm{Expm}(\tau H_{kl}).$$

We know that

$$\mathrm{Expm}(\tau H_{kl}) = I - (q_k q_k^\top + q_\ell q_\ell^\top) + \cos(\tau)(q_k q_k^\top + q_\ell q_\ell^\top) + \sin(\tau)(q_k q_\ell^\top - q_\ell q_k^\top),$$

and the formula for $U_+(\tau)$ follows from this. A similar argument holds for $V_+(\tau)$. $\qquad\square$

To prove Proposition 4.6, we need expressions for the Riemannian gradient. We derive these in Lemmas C.3 to C.4.

**Lemma C.3.** *Let $G = \nabla\widetilde{\widetilde{f}}(UPV^\top)$ be the Euclidean gradient. Then*

$$\nabla_U \bar{f}(U, P, V^\top) = GVP, \quad \nabla_P \bar{f}(U, P, V^\top) = U^\top GV, \quad \nabla_V \bar{f}(U, P, V^\top) = G^\top UP$$

*Proof of Lemma C.3.* Throughout the proof, denote $\langle\cdot,\cdot\rangle$ as the Euclidean inner product. Fix $U' \in \mathbb{R}^{m\times r}$, $V' \in \mathbb{R}^{n\times r}$ and $P' \in \mathbb{R}^{r\times r}$. Then

$$\langle\nabla_U \bar{f}(U, P, V^\top), U'\rangle = \frac{d}{dt}\widetilde{\widetilde{f}}((U+tU')PV^\top)\Big|_{t=0} = \frac{d}{dt}\widetilde{\widetilde{f}}(UPV^\top + tU'PV^\top)\Big|_{t=0} = \langle G, U'PV^\top\rangle$$

$$\langle\nabla_P \bar{f}(U, P, V^\top), P'\rangle = \frac{d}{dt}\widetilde{\widetilde{f}}(U(P+tP')V^\top)\Big|_{t=0} = \frac{d}{dt}\widetilde{\widetilde{f}}(UPV^\top + tUP'V^\top)\Big|_{t=0} = \langle G, UP'V^\top\rangle$$

$$\langle\nabla_V \bar{f}(U, P, V^\top), V'\rangle = \frac{d}{dt}\widetilde{\widetilde{f}}(UP(V+tV')^\top)\Big|_{t=0} = \frac{d}{dt}\widetilde{\widetilde{f}}(UPV^\top + tUP(V')^\top)\Big|_{t=0} = \langle G, UP(V')^\top\rangle.$$

The equalities then follow from these. $\qquad\square$

**Lemma C.4.** *The Riemannian gradient of $\widetilde{f}$ at $(U, P, V)$ is given by*

$$\nabla\widetilde{f}(U, P, V) = (U\tilde{A}_1 + U^\perp\tilde{B}_1, \tilde{D}, V\tilde{A}_2 + V^\perp\tilde{B}_2),$$

*where*

$$\tilde{A}_1 = U^\top GVP - PV^\top G^\top U, \quad \tilde{B}_1 = (U^\perp)^\top GVP,$$

$$\tilde{D} = \frac{1}{2}P(V^\top G^\top U + U^\top GV)P,$$

$$\tilde{A}_2 = V^\top G^\top UP - PU^\top GV, \quad \tilde{B}_2 = (V^\perp)^\top G^\top UP.$$

*Proof of Lemma C.4.* Let $\nabla\widetilde{f}(U, P, V)$ be the Riemannian gradient of $\widetilde{f}$ at $(U, P, V)$. We know that the Riemannian gradient satisfies

$$\mathrm{D}\widetilde{f}(U, P)[UA_1 + U^\perp B_1, D, VA_2 + V^\perp B_2] = \langle\nabla\widetilde{f}(U, P, V), (UA_1 + U^\perp B_1, D, VA_2 + V^\perp B_2)\rangle_{U,P,V}$$

for any tangent vector $(UA_1 + U^\perp B_1, D, VA_2 + V^\perp B_2) \in T_{U,P,V}\widetilde{M}_{m\times n,r}$. Furthermore, Boumal (2023, Eq. (3.36)) states that

$$\mathrm{D}\widetilde{f}(U, P, V)[UA_1 + U^\perp B_1, D, VA_2 + V^\perp B_2]$$
$$= \mathrm{D}\bar{f}(U, P, V)[UA_1 + U^\perp B_1, D, VA_2 + V^\perp B_2]$$
$$= \langle\nabla\bar{f}(U, P, V), (UA_1 + U^\perp B_1, D, VA_2 + V^\perp B_2)\rangle$$

with the Euclidean inner product. Now denoting $\nabla \widetilde{f}(U, P, V) = (U\tilde{A}_1 + U^\perp \tilde{B}_1, \tilde{D}, V\tilde{A}_2 + V^\perp \tilde{B}_2)$, we observe that

$$\langle \nabla \widetilde{f}(U, P, V), (UA_1 + U^\perp B_1, D, VA_2 + V^\perp B_2) \rangle_{U,P,V}$$

$$= \frac{1}{2}\langle \tilde{A}_1, A_1 \rangle + \langle \tilde{B}_1, B_1 \rangle + \langle P^{-1}\tilde{D}P^{-1}, D \rangle + \frac{1}{2}\langle \tilde{A}_2, A_2 \rangle + \langle \tilde{B}_2, B_2 \rangle$$

$$\langle \nabla \bar{f}(U, P, V), (UA_1 + U^\perp B_1, D, VA_2 + V^\perp B_2) \rangle$$

$$= \langle GVP, UA_1 + U^\perp B_1 \rangle + \langle U^\top GV, D \rangle + \langle G^\top UP, VA_2 + V^\perp B_2 \rangle.$$

These two terms must be equal for all $A_1, A_2 \in \mathrm{Skew}(r)$, $B_1 \in \mathbb{R}^{(m-r)\times r}$, $B_2 \in \mathbb{R}^{(n-r)\times r}$ and $D \in \mathbb{S}^r$. Note that $\mathrm{Skew}(r)^\perp = \mathbb{S}^r$ and vice versa, and $(\mathbb{R}^{(n-r)\times r})^\perp = \{0\}$. Therefore there exists $S_1, S_2 \in \mathbb{S}^r$ and $K \in \mathrm{Skew}(r)$ such that

$$\tilde{A}_1 = 2U^\top GVP + S_1$$

$$\tilde{B}_1 = (U^\perp)^\top GVP$$

$$P^{-1}\tilde{D}P^{-1} = U^\top GV + K$$

$$\tilde{A}_2 = V^\top G^\top UP + S_2$$

$$\tilde{B}_2 = (V^\perp)^\top G^\top UP$$

Since $\tilde{A}_1 \in \mathrm{Skew}(r)$, we have

$$0 = \tilde{A}_1 + \tilde{A}_1^\top = U^\top GVP + PV^\top G^\top U + 2S_1$$

$$\implies S_1 = -(U^\top GVP + PV^\top G^\top U)$$

$$\implies \tilde{A}_1 = U^\top GVP - PV^\top G^\top U.$$

Similarly, since $\tilde{A}_2 \in \mathrm{Skew}(r)$, we have

$$0 = \tilde{A}_2 + \tilde{A}_2^\top$$

$$= 2(V^\top G^\top UP + PU^\top GV) + 2S_2$$

$$\implies S_2 = -(V^\top G^\top UP + PU^\top GV)$$

$$\implies \tilde{A}_2 = V^\top G^\top UP - PU^\top GV.$$

Finally, $\tilde{D} \in \mathbb{S}^r$, we have

$$V^\top G^\top U + K^\top = P^{-1}\tilde{D}^\top P^{-1} = P^{-1}\tilde{D}P^{-1} = U^\top GV + K$$

$$\implies K = \frac{1}{2}(V^\top G^\top U - U^\top GV)$$

$$\implies P^{-1}\tilde{D}P^{-1} = \frac{1}{2}(V^\top G^\top U + U^\top GV).$$

Multiplying on the left and right by $P$ gives the result. $\qquad\square$

We need the following technical lemma to prove Proposition 4.6.

**Lemma C.5.** *Let $V$ be an inner product space with subspace $S \subset V$. Let $u \in V$. Then $\bar{v}$ is the projection of $u$ onto $S$ if and only if $u - \bar{v} \in S^\perp$.*

*Proof of Lemma C.5.* This follows by considering the optimality condition of minimizing $f(v) = \frac{1}{2}\langle u - v, u - v \rangle$ over $v \in S$, which is $\langle u - \bar{v}, v - \bar{v} \rangle \geq 0$ for all $v \in S$. Necessarily, we need $u - \bar{v} \in \bar{S}^\perp$. $\qquad\square$

We now state and prove Proposition 4.6 in full detail.

**Proposition 4.6** (Restated). *If $G$ is the Euclidean gradient of some extension of $f$ to all of $\mathbb{R}^{m \times n}$, then the cost of projecting $\nabla \widetilde{f}$ onto each subspace in the horizontal decomposition* (8) *is listed below by type:*

*Let $\nabla \widetilde{f}(U, P, V) = (U\tilde{A}_1 + U^\perp \tilde{B}_1, \tilde{D}, V\tilde{A}_2 + V^\perp \tilde{B}_2)$ as in Lemma C.4. Gradient projections onto subspaces in* (8) *are computed as follows:*

1. *Fix $k \in [r]$. We have $\mathrm{Proj}_{S_{1,k}(U,P,V)}(\nabla \widetilde{f}(U,P,V)) = (u q_k^\top, 0, 0)$ where $u \in \mathrm{Ker}(U^\top)$ is defined as*
$$u := \lambda_k(I - UU^\top)GVq_k.$$
   *Note also that $Vq_k$ is the kth column of $VQ$, and $UU^\top = UQ(UQ)^\top$.*

2. *Fix $k \in [r]$. We have $\mathrm{Proj}_{S_{2,k}(U,P,V)}(\nabla \widetilde{f}(U,P,V)) = (0, 0, v q_k^\top)$ where $v \in \mathrm{Ker}(V^\top)$ is defined as*
$$v := \lambda_k(I - VV^\top)G^\top Uq_k.$$
   *Note also that $Uq_k$ is the kth column of $VQ$, and $VV^\top = VQ(VQ)^\top$.*

3. *Fix $1 \le i \le j \le r$. Define $z_{ij}^P := (U(P^{-1}D_{ij}^P - D_{ij}^P P^{-1}), \frac{1}{2}D_{ij}^P, V(P^{-1}D_{ij}^P - D_{ij}^P P^{-1}))$. Then $S_{\mathbb{S},ij}(U, P, V) = \mathrm{Span}(\{z_{ij}^P\})$, hence*
$$\mathrm{Proj}_{S_{\mathbb{S},ij}(U,P,V)}(\nabla \widetilde{f}(U,P,V)) = \frac{\langle \nabla \widetilde{f}(U,P,V), z_{ij}^P \rangle_{U,P,V}}{\|z_{ij}^P\|_{U,P,V}^2} z_{ij}^P$$

   *where*
$$\langle \nabla \widetilde{f}(U,P,V), z_{ij}^P \rangle_{U,P,V} = \left(\sqrt{\frac{\lambda_i}{\lambda_j}} - \sqrt{\frac{\lambda_j}{\lambda_i}}\right)(\lambda_i + \lambda_j)\left((Uq_j)^\top GVq_i - (Uq_i)^\top GVq_j\right)$$
$$+ \sqrt{\lambda_i \lambda_j}\left((Uq_j)^\top GVq_i + (Uq_i)^\top GVq_j\right)$$
$$\|z_{ij}^P\|_{U,P,V}^2 = \frac{(\lambda_i - \lambda_j)^2}{\lambda_i \lambda_j} + \frac{1}{2}.$$

4. *Fix $1 \le k < \ell \le r$. Define $w_{k\ell}^P := (UH_{k\ell}, 0, -VH_{k\ell})$. Then $S_{\mathrm{Skew},ij}(U, P, V) = \mathrm{Span}(\{w_{k\ell}^P\})$, hence*
$$\mathrm{Proj}_{S_{\mathrm{Skew},ij}(U,P,V)}(\nabla \widetilde{f}(U,P,V)) = \frac{\langle \nabla \widetilde{f}(U,P,V), w_{k\ell}^P \rangle_{U,P,V}}{\|w_{k\ell}^P\|_{U,P,V}^2} w_{k\ell}^P$$
$$= \frac{1}{4}(\lambda_i + \lambda_j)\left((Uq_i)^\top GVq_j - (Uq_j)^\top GVq_i\right)w_{k\ell}^P.$$

*Proof of Proposition 4.6.*

1. Notice that $u = U^\perp \tilde{B}_1 q_k$. Furthermore, for any $u' \in \mathrm{Ker}(U^\top)$ we can write $u' = U^\perp w'$. Therefore we have
$$\langle \nabla f(U,P,V) - (u q_k^\top, 0, 0), (u' q_k^\top, 0, 0) \rangle_{U,P,V}$$
$$= \frac{1}{2}\langle \tilde{B}_1 - \tilde{B}_1 q_k q_k^\top, w' q_k^\top \rangle = 0.$$

   Lemma C.5 then states that $\left(U^\perp \tilde{B} q_k q_k^\top, 0\right)$ is exactly the projection.

2. The proof is similar to 1.

3. We have
$$\langle \nabla \widetilde{f}(U,P,V), z_{ij}^P \rangle_{U,P,V} = \frac{1}{2}\langle P^{-1}D_{ij}^P - D_{ij}^P P^{-1}, \tilde{A}_1 + \tilde{A}_2 \rangle$$
$$+ \frac{1}{2}\langle P^{-1/2}D_{ij}^P P^{-1/2}, P^{-1/2}\tilde{D}P^{-1/2} \rangle.$$

We can check that

$$\frac{1}{2}\langle P^{-1}D_{ij}^P - D_{ij}^P P^{-1}, \tilde{A}_1\rangle = \left(\sqrt{\frac{\lambda_i}{\lambda_j}} - \sqrt{\frac{\lambda_j}{\lambda_i}}\right)\left(\lambda_i(Uq_j)^\top GVq_i - \lambda_j(Uq_i)^\top GVq_j\right)$$

$$\frac{1}{2}\langle P^{-1}D_{ij}^P - D_{ij}^P P^{-1}, \tilde{A}_2\rangle = \left(\sqrt{\frac{\lambda_i}{\lambda_j}} - \sqrt{\frac{\lambda_j}{\lambda_i}}\right)\left(\lambda_j(Uq_j)^\top GVq_i - \lambda_i(Uq_i)^\top GVq_j\right).$$

Thus

$$\frac{1}{2}\langle P^{-1}D_{ij}^P - D_{ij}^P P^{-1}, \tilde{A}_1 + \tilde{A}_2\rangle = \left(\sqrt{\frac{\lambda_i}{\lambda_j}} - \sqrt{\frac{\lambda_j}{\lambda_i}}\right)(\lambda_i + \lambda_j)\left((Uq_j)^\top GVq_i - (Uq_i)^\top GVq_j\right).$$

Furthermore,

$$\frac{1}{2}\langle P^{-1/2}D_{ij}^P P^{-1/2}, P^{-1/2}\tilde{D}P^{-1/2}\rangle = \frac{1}{2}\langle D_{ij}^P, P^{-1}\tilde{D}P^{-1}\rangle$$

$$= \frac{1}{2}\langle D_{ij}^P, V^\top G^\top U + U^\top GV\rangle$$

$$= \sqrt{\lambda_i \lambda_j}\left((Uq_j)^\top GVq_i + (Uq_i)^\top GVq_j\right).$$

Finally, observe that $P^{-1}D_{ij}^P - D_{ij}^P P^{-1} = \sqrt{\lambda_i \lambda_j}\left(\frac{1}{\lambda_i} - \frac{1}{\lambda_j}\right)H_{ij}$ where $H_{ij} = q_i q_j^\top - q_j q_i^\top$, and $P^{-1/2}D_{ij}^P P^{-1//2} = S_{ij} = q_i q_j^\top + q_j q_i^\top$. Furthermore, $\langle H_{ij}, H_{ij}\rangle = \langle S_{ij}, S_{ij}\rangle = 2$ thus

$$\|z_{ij}^P\|_{U,P,V}^2 = \frac{(\lambda_i - \lambda_j)^2}{\lambda_i \lambda_j} + \frac{1}{2}.$$

4. The formula holds since

$$\langle\nabla\widetilde{f}(U,P,V), w_{k\ell}^P\rangle_{U,P,V} = \frac{1}{2}\langle H_{k\ell}, \tilde{A}_1 - \tilde{A}_2\rangle$$

$$= \frac{1}{2}\langle q_i q_j^\top - q_j q_i^\top, (U^\top GV - V^\top G^\top U)P - P(U^\top GV - V^\top G^\top U)\rangle$$

$$= (\lambda_i + \lambda_j)\left((Uq_i)^\top GVq_j - (Uq_j)^\top GVq_i\right)$$

and $\|w_{k\ell}^P\|_{U,P,V}^2 = 4$.

$\square$

# D    SUPPLEMENT TO SECTION 5

In this section, we use the notation $M = \mathbb{S}_+^{n,r}$ and $\widetilde{M}_{n,r,+} = \text{St}(n,r) \times \mathbb{S}_{++}^r$.

## D.1    RIEMANNIAN QUOTIENT STRUCTURE FOR THE FIXED-RANK PSD MATRIX MANIFOLD

This subsection focuses on the proof of Lemma 5.1, which establishes the Riemannian quotient structure on the manifold of fixed-rank positive semidefinite matrices, $M = \mathbb{S}_+^{n,r}$. We follow largely the same outline as for the fixed-rank matrix manifold, $\mathbb{R}_r^{m\times n}$.

We shall first classify the fibers of $\pi$, and exhibit a smooth group action of the $r \times r$-orthogonal group, $O(r)$, on $\mathbb{S}_+^{n,r}$ that is vertical and transitive on fibers. The proof follows quickly from the next lemma.

**Lemma D.1.** *The following hold for the fibers of $\pi$:*

1. *If $A \in \mathbb{S}_+^{n,r}$ has the singular value decomposition $UDU^\top$ then $\pi^{-1}(A) = \{(UW, W^\top DW) : W \in O(r)\}$.*

2. *$O(r)$, a Lie group, smooth and vertically acts on $\mathbb{S}_+^{n,r}$ as $(U,P) \cdot W := (UW, W^\top PW)$.*

3. $O(r)$ acts transitively on fibers, and each fiber is properly embedded submanifold diffeomorphic to $O(r)$.

*Proof of Lemma D.1.* We shall prove each item separately.

1. The inclusion $\{(UW, W^\top DW) : W \in O(r)\} \subseteq \pi^{-1}(A)$ is an immediate consequence of matrix multiplication, so we only prove the reverse inclusion. Choose $(\widetilde{U}, P) \in \widetilde{M}_{n,r,+}$ such that $\widetilde{U} P \widetilde{U}^\top = A$. Let $W = U^\top \widetilde{U}$. By Lemma C.2, with $V = U$ we can find some $W \in O(r)$ such that $\widetilde{U} = UW$, $P = W^\top DW$.

2. The action of $O(r)$ on $\mathbb{R}^{m \times r} \times \mathbb{R}^{r \times r}$ defined as $(U, P) \cdot W := (UW, W^\top P)$ is readily seen to be smooth because it is defined by matrix multiplications and transpositions. Being a product of embedded submanifolds of $\mathbb{R}^{m \times r} \times \mathbb{R}^{r \times r}$, $\widetilde{M}_{n,r,+}$ is an embedded submanifold. Thus, $O(r)$ restricts to a smooth action on this set. By the first item, this action is vertical.

3. By the first item, $O(r)$ is transitive on fibers. For a fixed $(U, P, V) \in \widetilde{M}_{n,r,+}$, the map $W \in O(r) \mapsto (UW, W^\top PW)$ is bijective and smooth, so it is an immersion by (Lee, 2012, Proposition 7.26). AS $O(r)$ is compact, this map is smooth embedding by (Lee, 2012, Proposition 4.22), and by (Lee, 2012, Corollary 5.6) it is proper embedding. □

Now, we are fully equipped to prove Lemma 5.1 regarding the Riemannian quotient structure for $\mathbb{R}^{m \times n, r}$.

**Lemma 5.1** (Riemannian Quotient Structure for $\mathbb{S}_+^{n,r}$). *The map $\pi$ is a Riemannian quotient map when $\mathbb{S}_+^{n,r}$ is endowed with its embedded smooth structure and a suitable Riemannian metric. Additionally, the vertical space at $(U, P) \in \widetilde{M}_{n,r,+}$ is given by*

$$\mathcal{V}_{(U,P)} = \{(UA, PA - AP) : A \in \mathrm{Skew}(r)\}.$$

*Proof of Lemma 5.1.* We follow a two step process:

1. The map $\pi$ is a smooth submersion. This requires only that we show the vertical space at $(U, P) \in \widetilde{M}_{n,r,+}$ is given by

$$\mathcal{V}_{U,P,V} = \{(UA, PA - AP) : A \in \mathrm{Skew}(r)\},$$

because the vertical spaces have the same dimension across $\widetilde{M}_{n,r,+}$. The map $\pi$ is surjective so the Global Rank Theorem (Lee, 2012, Theorem 4.14) implies $\pi$ is a smooth submersion.

2. The group $O(r)$ acts isometrically on $\widetilde{M}_{n,r,+}$ given the metric (9b). This fact, in conjunction with Lemma D.1 and Lee (2012, Theorem 2.28), implies that $\mathbb{R}_r^{m \times}$ can be endowed with a metric making $\pi$ is a Riemannian quotient map.

1. Fix $(U, P) \in M_{m \times n, r}$. Let

$$\tilde{\mathcal{V}}_{(U,P)} = \{(UA, PA - AP) : A \in \mathrm{Skew}(r)\}.$$

We need to show that $\tilde{\mathcal{V}}_{(U,P)} = \mathcal{V}_{(U,P)}$. The inclusion, $\tilde{\mathcal{V}}_{(U,P)} \subseteq \mathcal{V}_{(U,P)}$, is relatively straightforward. The differential of the diffeomorphism $W \in O(r) \mapsto (UW, W^\top PW)$ at $I_r$ is

$$H \mapsto (UH, PH - HP)$$

for $H \in T_{I_r} O(r) = \mathrm{Skew}(r)$. As the map is a diffeomorphism, the differential is an isomorphism onto $T_{(U,P)} \pi^{-1}(UPU^\top) = \mathrm{Ker}(d\pi_{(U,P)})$, and the inclusion follows.

We move onto the next inclusion. Suppose that $(H, D) \in T_{(U,P,V)} M_{m \times n, r}$ such that $d\pi_{(U,P)}(H, D) = 0$. Consider a curve $t \mapsto (U + tH, P + tD)$, and the corresponding curve

$$t \mapsto \pi(U + tH, P + tD) = (U + tH)(P + tD)(U + tH)^\top.$$

We are permitted to consider these curves in computing $\ker(d\pi_{(U,P)})$ because $\widetilde{M}_{n,r,+}$ and $\mathbb{S}^{n,r}_+$ are properly embedded submanifolds of their ambient Euclidean spaces. These proper embeddings allow us to consider $\pi$'s extension to the entire ambient space of $\widetilde{M}_{n,r,+}$, which has codomain $\mathbb{S}^n$, when computing $\ker(d\pi_{(U,P)})$. Taking derivatives at $t = 0$, we see $d\pi_{(U,P)}(U + tH, P + tD) = \frac{d}{dt}\pi(U + tH_1, P + tD)|_{t=0}$, which produces

$$d\pi_{(U,P)}(H, D) = HPU^\top + UDU^\top + UPH^\top.$$

There exist $A \in \text{Skew}(r)$, $B \in \mathbb{R}^{(n-r)\times r}$ such that $H = UA + U^\perp B$. Plugging these into the above equation, we find the condition

$$0 = d\pi_{(U,P)}(H, D) = (UA + U^\perp B)PU^\top + UDU^\top + UP(UA + U^\perp B)^\top$$
$$= U(AP + PA^\top + D)U^\top + U^\perp BPU^\top + UPB^\top(U^\perp)^\top$$

Multiplying on the left and the right by $U^\top$ and $U$ respectively, we obtain $D = PA - AP$. Multiplying on the left by $U^\top P^{-1}$ and on the right by $U^\perp$, we obtain $B^\top = 0$. We thus, conclude that $(H, D)$ has the requisite form to belong to $\widetilde{\mathcal{V}}_{(U,P,V)}$.

2. We need only show that $O(r)$ acts isometrically on $M$ given the metric (7c). The metric is product metric so it suffices to show each of the constituent metrics are invariant under $O(r)$'s action on each component. This was established in the course of proving Lemma 4.1. $\qquad\square$

### D.2 HORIZONTAL SUBSPACE DECOMPOSITION

This subsection focuses on the proofs of Lemma 5.2, Proposition 5.3, and Proposition 5.4.

**Lemma 5.2** (Horizontal Space of $\mathbb{S}^{n,r}_+$). *For any $(U, P) \in \widetilde{M}_{n,r,+}$, we have*

$$\mathcal{H}_{(U,P)} = \left\{ \left( U(P^{-1}D - DP^{-1}) + U^\perp B, \frac{1}{2}D \right) : B \in \mathbb{R}^{(n-r)\times r}, D \in \mathbb{S}^r \right\}.$$

*Proof of Lemma 5.2.* Choose an arbitrary $(U\bar{A} + U^\perp \bar{B}, \bar{D}) \in T_{(U,P)}\widetilde{M}_{n,r,+}$. We shall find conditions on $\bar{A}, \bar{B}, \bar{D}$ so that

$$\left\langle (U\bar{A} + U^\perp \bar{B}, \bar{D}), (UA, PA - AP) \right\rangle_{U,P} = 0$$

for all $A \in \text{Skew}(r)$. The inner product can be written as

$$\left\langle (U\bar{A} + U^\perp \bar{B}, \bar{D}), (UA, PA - AP) \right\rangle_{U,P} = \frac{1}{2}\langle \bar{A}, A \rangle + \langle P^{-1}\bar{D}P^{-1}, PA - AP \rangle$$
$$= \frac{1}{2}\langle \bar{A}, A \rangle + \langle \bar{D}P^{-1}, A \rangle - \langle P^{-1}\bar{D}, A \rangle$$
$$= \left\langle \frac{1}{2}\bar{A} - \left( P^{-1}\bar{D} - \bar{D}P^{-1} \right), A \right\rangle.$$

Therefore the condition we need is

$$\frac{1}{2}\bar{A} - \left( P^{-1}\bar{D} - \bar{D}P^{-1} \right) \in \mathbb{S}^r.$$

But both matrices above are skew-symmetric, and thus orthogonal to $\mathbb{S}^r$, hence we have $\bar{A} = 2(P^{-1}\bar{D} - \bar{D}P^{-1})$. $\qquad\square$

**Proposition 5.3** (Horizontal Decomposition for $\mathbb{S}^{n,r}_+$). *Let $(U, P, V) \in \widetilde{M}_{n,r,+}$ and $P = Q\Lambda Q^\top$ be a spectral decomposition of $P$. The collection of horizontal subspaces*

$$S_k(U, P) := \left\{ (vq_k^\top, 0) : v \in \ker(U^\top) \right\}$$
$$S_{ij}(U, P) := \left\{ \tau \left( U(P^{-1}D_{ij}^P - D_{ij}^P P^{-1}), \frac{1}{2}D_{ij}^P \right) : \tau \in \mathbb{R} \right\}$$

*where $1 \le i \le j \le r$ and $1 \le k \le r$, forms an orthogonal decomposition of $T_{(U,P)}\widetilde{M}_{n,r,+}$ under the metric* (9b). *Moreover, there are $m = \frac{r(r+3)}{2}$ different subspaces in total, so the subspace selection rule that chooses uniformly from this collection satisfies Assumption 3.2 with $C = \frac{1}{\sqrt{m}} = \sqrt{\frac{2}{r(r+3)}}$.*

*Proof of Proposition 5.3.* It is clear that $S_k(U,P)$ and $S_{ij}(U,P)$ are orthogonal, and that $\{S_k(U,P) : k \in [r]\}$ are mutually orthogonal. We will prove that $S_{ij}(U,P)$ and $S_{kl}(U,P)$ are orthogonal whenever $(i,j) \neq (k,l)$.

To see this, observe that

$$\left\langle \left( U(P^{-1}D_{ij}^P - D_{ij}^P P^{-1}), \frac{1}{2}D_{ij}^P \right), \left( U(P^{-1}D_{kl}^P - D_{kl}^P P^{-1}), \frac{1}{2}D_{kl}^P \right) \right\rangle_{U,P}$$

$$= \frac{1}{2} \operatorname{Tr} \left( \left(P^{-1}D_{ij}^P - D_{ij}^P P^{-1}\right)^\top \left(P^{-1}D_{kl}^P - D_{kl}^P P^{-1}\right) \right)$$

$$+ \frac{1}{4} \operatorname{Tr} \left( \left(P^{-1/2}D_{ij}^P P^{-1/2}\right)^\top \left(P^{-1/2}D_{kl}^P P^{-1/2}\right) \right).$$

Denote $S_{ij} = q_i q_j^\top + q_j q_i^\top$, and analogously for $S_{kl}$.

For the second term, notice that by definition, we have $P^{-1/2}D_{ij}^P P^{-1/2} = S_{ij}$, and analogously for $P^{-1/2}D_{kl}^P P^{-1/2}$. Therefore

$$\operatorname{Tr} \left( \left(P^{-1/2}D_{ij}^P P^{-1/2}\right)^\top \left(P^{-1/2}D_{kl}^P P^{-1/2}\right) \right) = \operatorname{Tr}(S_{ij}S_{kl}).$$

This is 0 unless $(i,j) = (k,l)$ or $(i,j) = (l,k)$.

For the first term, notice that $P^{-1}D_{ij}^P - D_{ij}^P P^{-1} = (\lambda_i \lambda_i)^{1/2}(P^{-1}S_{ij} - S_{ij}P^{-1}) = (\lambda_i \lambda_i)^{1/2}(\lambda_i^{-1}q_i q_j^\top + \lambda_j^{-1}q_j q_i^\top - \lambda_j^{-1}q_i q_j^\top - \lambda_i^{-1}q_j q_i^\top) = (\lambda_i \lambda_j)^{1/2}(\lambda_i^{-1} - \lambda_j^{-1})(q_i q_j^\top - q_j q_i^\top)$.
Denote $H_{ij} = q_i q_j^\top - q_j q_i^\top$ and analogously for $H_{kl}$. Therefore

$$\left(P^{-1}D_{ij}^P - D_{ij}^P P^{-1}\right)^\top \left(P^{-1}D_{kl}^P - D_{kl}^P P^{-1}\right)$$

$$= (\lambda_i \lambda_j)^{1/2}(\lambda_i^{-1} - \lambda_j^{-1})(\lambda_k \lambda_l)^{1/2}(\lambda_k^{-1} - \lambda_l^{-1})H_{ij}^\top H_{kl}.$$

Therefore

$$\operatorname{Tr} \left( \left(P^{-1}D_{ij}^P - D_{ij}^P P^{-1}\right)^\top \left(P^{-1}D_{kl}^P - D_{kl}^P P^{-1}\right) \right) = (\lambda_i \lambda_j)^{1/2}(\lambda_i^{-1} - \lambda_j^{-1})(\lambda_k \lambda_l)^{1/2}(\lambda_k^{-1} - \lambda_l^{-1})\operatorname{Tr}(H_{ij}^\top H_{kl}).$$

This is equal to 0 unless $(i,j) = (k,l)$ or $(i,j) = (l,k)$.

Therefore, combining both cases,

$$\left\langle \left( U(P^{-1}D_{ij}^P - D_{ij}^P P^{-1}), \frac{1}{2}D_{ij}^P \right), \left( U(P^{-1}D_{kl}^P - D_{kl}^P P^{-1}), \frac{1}{2}D_{kl}^P \right) \right\rangle_{U,P} = 0$$

unless $(i,j) = (k,l)$ or $(i,j) = (l,k)$. However, since $i \leq j$, $k \leq l$, this only occurs if $i = k$, $j = l$. Therefore the subspaces $S_{ij}(U,P)$ are mutually orthogonal.

To see why Assumption 3.2 is satisfied, notice that the subspaces span $\mathrm{H}_{U,P}$, so

$$\mathrm{H}_{U,P} = \left( \bigoplus_{k \in [r]} S_k(U,P) \right) \oplus \left( \bigoplus_{1 \leq i \leq j \leq r} S_{ij}(U,P) \right),$$

and they are orthogonal. Therefore for any $v \in \mathrm{H}_{U,P}$, we have

$$\|v\|_{U,P} = \sqrt{\sum_{k \in [r]} \left\| \operatorname{Proj}_{S_k(U,P)}(v) \right\|_{U,P}^2 + \sum_{1 \leq i \leq j \leq r} \left\| \operatorname{Proj}_{S_{ij}(U,P)}(v) \right\|_{U,P}^2}$$

$$= \sqrt{\frac{r(r+3)}{2} \left( \sum_{k \in [r]} \frac{2}{r(r+3)} \left\| \operatorname{Proj}_{S_k(U,P)}(v) \right\|_{U,P}^2 + \sum_{1 \leq i \leq j \leq r} \frac{2}{r(r+3)} \left\| \operatorname{Proj}_{S_{ij}(U,P)}(v) \right\|_{U,P}^2 \right)}$$

$$= \sqrt{\frac{r(r+3)}{2} \mathbb{E} \left[ \left\| \operatorname{Proj}_{S(U,P)}(v) \right\|_{U,P}^2 \mid U, P \right]},$$

thus Assumption 3.2 is satisfied with $C = \frac{1}{\sqrt{m}} = \sqrt{\frac{2}{r(r+3)}}$. $\qquad\square$

We conclude with a proof of non-smoothness of the horizontal subspace decomposition.

**Proposition 5.4** (Non-smooth Randomized Subspace Selection for $\mathbb{S}_+^{n,r}$)**.** *If $r > 2$, then the subspace selection rule induced by the horizontal decomposition in Proposition 5.3 is not smooth.*

*Proof of Proposition 5.4.* It suffices to show a single subspace from the decomposition is non-smooth. We consider $S_{12}(U, P)$ which is spanned by

$$W(U, P) = \left( \left( \frac{1}{\lambda_1} - \frac{1}{\lambda_2} \right) U H_{12}, \frac{1}{2}(q_1 q_2^\top + q_2 q_1^\top) \right).$$

where $q_1$ and $q_2$ are eigenvectors with corresponding eigenvalues $\lambda_1$ and $\lambda_2$. In fact, as $S_{12}(U, P)$ has dimension equal to 1, any spanning vector of it must be a scalar multiple of $W(U, P)$. Set

$$U_0 = \begin{bmatrix} I_r \\ 0 \end{bmatrix}, \quad P_0 = I_r,$$

$q_1 = e_1$, and $q_2 = e_2$. Then

$$W(U_0, P_0) = \left( 0, \frac{1}{2}(e_1 e_2^\top + e_2 e_1^\top) \right).$$

Observe the eigenvectors for $P + \epsilon(e_1 e_2^\top + e_2 e_1^\top)$ are

$$\frac{1}{\sqrt{2}}(e_1 + e_2), \frac{1}{\sqrt{2}}(e_1 - e_2), e_3, \dots, e_r$$

and the eigenvectors for $P + \epsilon(e_1 e_3^\top + e_3 e_1^\top)$ are

$$\frac{1}{\sqrt{2}}(e_1 + e_3), \frac{1}{\sqrt{2}}(e_1 - e_3), e_2, e_4, \dots, e_r$$

Consequently, there is no selection of eigenvectors of $P$ that ensures $q_1$ and $q_2$ are continuous, $\lim_{P \to I_r} q_1(P) = e_1$, and $\lim_{P \to I_r} q_2(P) = e_2$. Thus, $W$ is discontinuous at $(U_0, P_0)$. $\square$

### D.3 EFFICIENT UPDATE FORMULAS: EXPONENTIAL MAPS & GRADIENT PROJECTIONS

In the main text, we presented streamlined statements for Theorem 5.5 and Proposition 5.6. In this section, we dive into the efficient expressions for the exponential map and the gradient projections. We provide complete statements of these results. As the proofs are very similar to Theorem 4.5 and Proposition 4.6, we provide proof outlines for brevity.

**Theorem 5.5** (Restated)**.** *Fix $\tau \in \mathbb{R}$. Let $(U, P) \in \widetilde{M}_{n,r,+}$ with $P = Q \Lambda Q^\top$ being a spectral decomposition for $P$, where $Q = [q_1 \quad \cdots \quad q_r] \in O(r), \Lambda = \mathrm{Diag}(\lambda_1, \dots, \lambda_r) \succ 0$. Let $W \in T_{(U,P)}\widetilde{M}_{n,r,+}$, and for $\tau \in \mathbb{R}$, we define $U_+(\tau), P_+(\tau), Q_+(\tau), \Lambda_+(\tau)$ as follows:*

$$(U_+(\tau), P_+(\tau)) = \widetilde{\mathrm{Exp}}_{(U,P)}(\tau W), \quad P_+(\tau) = Q_+(\tau)\Lambda_+(\tau)Q_+^\top(\tau).$$

*Then we have the following update rules:*

    *1. Fix any $k \in [r]$ and $u \in \ker(U^\top)$. If $W = (uq_k^\top, 0, 0) \in S_k(U, P)$, then*

$$U_+(\tau) = U(I - q_k q_k^\top) + \left( \cos(\tau\|u\|_2)Uq_k + \sin(\tau\|u\|_2)\frac{u}{\|u\|_2} \right) q_k^\top,$$

    *while $(P_+(\tau), Q_+(\tau), \Lambda_+(\tau)) = (P, Q, \Lambda)$. This means that $U_+(\tau)Q_+(\tau)$ is obtained by replacing the kth column of $UQ$, which is $Uq_k$, with a linear combination $\cos(\tau\|u\|_2)Uq_k + \sin(\tau\|u\|_2)\frac{u}{\|u\|_2}$.*

    *2. Fix any $1 \le i < j \le r$. Suppose $W = (U(P^{-1}D_{ij}^P - D_{ij}^P P^{-1}), \frac{1}{2}D_{ij}^P)) \in S_{\mathbb{S},ij}(U, P, V)$. Let $\alpha_{ij}(\tau) := \tau(\lambda_i \lambda_j)^{1/2}(\lambda_i^{-1} - \lambda_j^{-1})$. Then*

$$U_+(\tau) = U(I - q_i q_i^\top - q_j q_j^\top) + (\cos(\alpha_{ij}(\tau))Uq_i - \sin(\alpha_{ij}(\tau))Uq_j)q_i^\top$$
$$+ (\sin(\alpha_{ij}(\tau))Uq_i + \cos(\alpha_{ij}(\tau))Uq_j)q_j^\top$$
$$P_+(\tau) = P - (\lambda_i q_i q_i^\top + \lambda_j q_j q_j^\top)$$
$$+ \cosh(\tau/2)(\lambda_i q_i q_i^\top + \lambda_j q_j q_j^\top) + \sinh(\tau/2)\sqrt{\lambda_i \lambda_j}(q_i q_j^\top + q_j q_i^\top).$$

*Define*

$$M_{ij}(\tau) := \begin{bmatrix} \cosh(\tau/2)\lambda_i & \sinh(\tau/2)\sqrt{\lambda_i \lambda_j} \\ \sinh(\tau/2)\sqrt{\lambda_i \lambda_j} & \cosh(\tau/2)\lambda_j \end{bmatrix}, \quad G(\tau) := \begin{bmatrix} \cos(\alpha_{ij}(\tau)) & \sin(\alpha_{ij}(\tau)) \\ -\sin(\alpha_{ij}(\tau)) & \cos(\alpha_{ij}(\tau)) \end{bmatrix}.$$

*Let $e_1(\tau), e_2(\tau) \in \mathbb{R}^2$ be the two eigenvectors of $M_{ij}(\tau)$ with corresponding eigenvalues $\gamma_1(\tau), \gamma_2(\tau)$. Define $Q_{ij} := [q_i \quad q_j]$. Only the $i$ and $j$ columns of $U_+(\tau)Q_+(\tau)$; all others remain the same as those of $UQ$. These columns are updated as follows:*

$$Uq_i \to UQ_{ij}G(\tau)e_1(\tau), \quad Uq_j \to UQ_{ij}G(\tau)e_2(\tau).$$

*(Here, $UQ_{ij}$ is an $m \times 2$ matrix consisting of the $i$ and $j$ columns of $UQ$, and similarly for $VQ_{ij}$.) Furthermore, in $\Lambda_+(\tau)$, the entry $\lambda_i$ is replaced with $\gamma_1(\tau)$, while $\lambda_j$ is replaced with $\gamma_2(\tau)$, and all other entries remain the same.*

3. *Fix any $i \in [r]$. If $W = \left(U(P^{-1}D_{ii}^P - D_{ii}^P P^{-1}), \frac{1}{2}D_{ii}^P\right) \in S_{\mathbb{S},ii}(U, P)$, then $W = (0, D_{ii}^P/2)$. Thus*

$$P_+(\tau) = P - \lambda_i q_i q_i^\top + \lambda_i \exp(\tau) q_i q_i^\top.$$

*while $(U_+(\tau), Q_+(\tau)) = (U, Q)$. This means that $\lambda_i$ is replaced with $\lambda_i \exp(\tau)$ in $\Lambda_+$, while all other entries are the same.*

*Proof outline of Theorem 5.5.* The proof is almost identical to the corresponding cases (1, 3, 4) in Theorem 4.5. $\qquad\square$

Proposition 5.6 can be proven using the same techniques as those used to derive Proposition 4.6. We state the explicit formulas below, and provide an outline of the proof.

**Proposition 5.6** (Restated). *Suppose the following matrices are stored for a given $(U, P) \in \widetilde{M}_{n,r,+}$:*

1. *a spectral decomposition $P = Q\Lambda Q^\top$ for $P$ along with $UQ$ and $VQ$,*

2. *and the (Euclidean) gradient, $G := \nabla\bar{f}(UPV^\top) \in \mathbb{R}^{n \times n}$, where $\bar{f}$ is a Euclidean extension of $f : \mathbb{S}_+^{n,r} \to \mathbb{R}$.*

*Then the Riemannian gradient is*

$$\nabla\widetilde{f}(U, P) = (U\tilde{A} + U^\perp\tilde{B}, \tilde{D})$$
$$\text{where } \tilde{A} = U^\top(G + G^\top)UP - PU^\top(G + G^\top)U$$
$$\tilde{B} = (U^\perp)^\top(G + G^\top)UP$$
$$\tilde{D} = \frac{1}{2}PU^\top(G + G^\top)UP.$$

*Furthermore, projections of $\nabla\widetilde{f}(U, P)$ onto subspaces in (10) are computed as follows:*

$$\text{Proj}_{\mathcal{S}_k^{U,P}}(\nabla\widetilde{f}(U, P)) = \left(U^\perp\tilde{B}q_k q_k^\top, 0\right) = \left(\lambda_k(I_n - UU^\top)(G + G^\top)Uq_k q_k^\top, 0\right)$$

$$\text{Proj}_{\mathcal{S}_{ij}^{U,P}}(\nabla\widetilde{f}(U, P)) = \frac{\langle\nabla\widetilde{f}(U, P), v_{ij}^P\rangle_{U,P}}{\|v_{ij}^P\|_{U,P}^2}v_{ij}^P = \sqrt{\lambda_i \lambda_j}(Uq_j)^\top(G + G^\top)(Uq_i)v_{ij}^P$$

$$\text{where } v_{ij}^P := \left(U(P^{-1}D_{ij}^P - D_{ij}^P P^{-1}), \frac{1}{2}D_{ij}^P\right).$$

*Proof outline of Proposition 5.6.* The Riemannian gradient derivation proceeds by first deriving the Euclidean gradient of $\bar{f}$, similar to Lemma C.3. Then, the Riemannian gradient expression is obtained through Boumal (2023, Eq. (3.36)) and performing some algebraic arguments, as in Lemma C.4. Projections are computed by again employing Lemma C.5, and then performing the necessary algebraic simplifications. $\square$

