# OpenReview forum: "Randomized Subspace Methods for Optimization on Fixed-Rank Matrix Manifolds"
_ICLR.cc/2026/Conference — Submitted to ICLR 2026_

### Official Review · Reviewer_HkdT · 2025-10-29

**Soundness:** 3
**Presentation:** 2
**Contribution:** 2
**Rating:** 4
**Confidence:** 2

**Summary:**

This paper proposes an efficient optimization method that uses randomized subspace descent to optimize functions over fixed-rank matrix manifolds. The representation of the low-rank matrix is lifted to a product manifold with simpler geometry, and the subspaces are aware of the quotient structure. The numerical computations are carefully designed, so that the computation of exponential maps are minimized to low-cost operations such as the rank-one updates. Experiment on a toy problem preliminarily verifies the effectiveness compared to the full Riemannian gradient descent.

**Strengths:**

1. originality. The most interesting part of the work is the decomposition of the horizontal subspace. The decomposition is carefully constructed so that the update with exponential map has low-cost. However, the idea of lifting the problem to a Riemannian product manifold is not new, and is widely used ([a, b]).

2. quality and clarity. The presentation is clear, with sufficient discussion on technical preliminaries and backgrounds. The difference of the proposed method with previous work is also highlighted.

3. significance. The low-rank matrices are important objects to study, and they are also having applications in LLMs in recent years. Having more efficient way of optimizing them will benefit downstream applications.

- [a] R3MC: A Riemannian three-factor algorithm for low-rank matrix completion, 53rd IEEE Conference on Decision and Control, 2014
- [b] StelLA: Subspace Learning in Low-rank Adaptation using Stiefel Manifold, NeurIPS 2025

**Weaknesses:**

1. The benefit of non-smooth subspace v.s. the smooth version is not justified. I do not understand why this difference is highlighted so much (e.g., Thm 3.1).
2. The numerical study is limited. No verification on low-rank matrices, which is the main topic of the paper. No comparison to other TSD based methods. Maybe it is better to shorten Sec 2 or 3 or 5, and enhance Sec 6.
3. No discussion about limitation; No conclusion section; The texts in Fig. 1 are too small to read.

**Questions:**

1. What if the product space of Eq. (7a) is simply $St(m, r) \times \mathbb{R}^r \times St(n, r)$, where the middle factor could be $r$ positive numbers, and call a diag() function when using it. This way, maybe the whole complexity involving positive definite matrices are gone.
2. In Eq. (8a), how $u$ is selected? What is the complexity?
3. Can this method be extended to work with mometum, like in SGD? Can it be integrated to deep learning frameworks (which is easy for the full RGD, e.g. in [b]), like PyTorch?
4. In Sec. 6.1, what's the complexity of projecting gradients if all the Euclidean gradients are accessible? This scenario is true when an auto-differentiation framework is used. In this case, the comparision is no longer problem-dependent.

---

### Official Review · Reviewer_HHwS · 2025-10-31

**Soundness:** 3
**Presentation:** 3
**Contribution:** 3
**Rating:** 4
**Confidence:** 3

**Summary:**

The paper introduces randomized subspace methods for optimization on fixed-rank matrix manifolds, covering both the manifold of fixed-rank rectangular matrices and the manifold of rank $r$ positive semidefinite matrices. The key idea is to formulate these manifolds as Riemannian quotient manifolds of convenient product spaces, and then to design Tangent Subspace Descent (TSD) algorithms that operate in the horizontal space induced by the quotient structure.

Two technical innovations underpin the approach. 1) Non-smooth subspace selection: Unlike prior TSD/coordinate methods whose selectable tangent subspaces vary smoothly across the manifold, the proposed horizontal subspace families are not smooth distributions. The paper formalizes this distinction and proves non-smoothness. 2) Lifting via Riemannian quotients: By working in the total (product) space, the authors obtain simple, low-cost expressions for projections of gradients onto horizontal subspaces and for exponential map updates. Crucially, each update requires at most rank one modifications on Stiefel parts and at most a $2\times 2$ eigendecomposition for SPD parts, with minimal state $(UQ, VQ, diag(\Lambda))$ stored.

Within the TSD framework, the authors provide: i) Quotient geometry, vertical/horizontal space characterizations, and horizontal subspace decompositions that satisfy the $C$-randomized norm assumption needed for convergence; ii) Efficient algorithms for exponential maps and gradient projections on the selected subspaces with $O(n)$ or $O(r)$ per-step costs on the Stiefel/SPD components, and $O(mn)$ worst-case for projections (often much less for structured problems); iii) A proof-of-concept empirical study on trace regression over $S_+^{n,r}$, showing that the proposed randomized TSD closes optimization gaps faster than Riemannian gradient descent (RGD) in time and cycles, with the advantage diminishing somewhat as $r$ increases.

Overall, the paper establishes a practical randomized subspace methodology for fixed-rank matrix manifolds, with substantial per-iteration efficiency gains.

**Strengths:**

Originality: 1) First randomized subspace (coordinate-like) methods tailored to fixed-rank matrix manifolds via quotient lifting. This goes beyond prior TSD instantiations (orthogonal/Stiefel/PD) by handling quotient geometry and by introducing non-smooth subspace selection. 2) The decomposition of the horizontal space into orthogonal subspaces tied to spectral directions of the SPD core is elegant and appears novel; the non-smoothness result is interesting and clarifies a conceptual gap with earlier smooth-selection approaches.

Quality: 1) Solid geometric foundations: clear quotient constructions, vertical/horizontal space characterizations, and proofs that the selection rules satisfy the C-randomized norm assumption for convergence guarantees under TSD. 2) Implementation-relevant derivations: closed-form exponential-map updates per subspace (only $2×2$ diagonalizations, rank one updates, or simple sines/cosines), and explicit projected-gradient formulas with tight complexity bounds. 3) Empirical validation on a representative application (trace regression) demonstrates practical gains over RGD, consistent with the promised computational savings.

Clarity: 1) The paper is well organized: preliminaries, framework, then separate sections for $R_r^{m×n}$ and $S_+^{n,r}$, each delivering the three required ingredients (quotient geometry, horizontal decomposition, efficient updates). 2) Algorithmic roles of stored quantities $(UQ, VQ, \Lambda)$ are made explicit; appendices provide complete formulas and proofs for reproducibility. 3) The discussion distinguishes clearly between smooth vs. non-smooth subspace selection, and why non-smoothness arises here (eigenvector discontinuities).

Significance: 1) Fixed-rank constrained optimization is central to matrix completion, compressed sensing, semidefinite approximation, and beyond. Lower per-iteration costs with preserved first-order iteration complexity is impactful. 2) The quotient-lifted TSD approach offers a pathway for scalable first-order methods on low-rank manifolds that avoid expensive full decompositions or generic retractions, with potential to influence future randomized/block methods in Riemannian optimization.

**Weaknesses:**

Empirical scope and baselines: 1) Experiments focus on trace regression with synthetic data and compare only against vanilla Riemannian gradient descent. Stronger baselines (e.g., Riemannian conjugate gradient, stochastic/variance-reduced RO, recent retraction-based randomized subspace methods) are absent. It is hard to gauge how much of the speedup is due to subspace randomization vs. quotient-engineered cheap updates versus baseline implementation choices. 2) Limited range of problem sizes and ranks ($n$ at most $50$); no real-world datasets (e.g., matrix completion, low-rank SDP approximations) where structure could further amplify the proposed projection savings.

Practical guidance on subspace sampling: 1) The selection rule is uniform over $\frac{r(2r+3)}{2}$ subspaces. While sufficient for theory, there’s little discussion on importance sampling or adaptive probabilities (e.g., based on gradient energy in subspaces) that could improve convergence speed in practice. 2) Sensitivity and numerical stability: i) The non-smooth selection hinges on eigendecompositions of $P$ and uses eigenvector-based subspaces. Near-multiplicities or clustered eigenvalues can cause eigenvector instability; while the updates handle only small $2×2$ blocks, the paper does not empirically analyze robustness or propose tie-breaking/regularization strategies for clustered spectra. ii) Storage scheme assumes maintaining $UQ$, $VQ$, and $\Lambda$; however, orthogonality drift, accumulation of rounding errors, and the cost/frequency of re-orthogonalization are not discussed.

Convergence details: The global guarantees hinge on the TSD framework with smoothness and $C$-randomized norm assumptions for the lifted function. Practical step-size choices rely on Armijo line search in experiments, but there’s no guidance on step-size tuning or bounds for Lipschitz constants $\tilde{L}$ in realistic settings.

**Questions:**

The experimental results are not convincing. 1) Could you include comparisons with stronger RO baselines such as Riemannian conjugate gradient, trust-region methods, or recent retraction-based randomized subspace methods on product manifolds? This would contextualize gains beyond simple RGD. 2) Please add experiments on large-scale problems. The current experiments is for $n\leq 50$. Please also add experiments on matrix completion or semidefinite approximation with real datasets, where your projection shortcuts can exploit structure and where fixed-rank PSD models are common.

Eigenvalue clustering and robustness: 1) How does the method behave when $P$ has repeated or tightly clustered eigenvalues? Do you observe instability in $UQ$ $VQ$ updates due to eigenvector switching? Any safeguards (e.g., subspace rotations, block updates on eigenspaces, or regularized alignment) to mitigate non-smoothness in practice? 2) It would help to provide an ablation where the condition number of $P$ and spectral gaps are varied, reporting iteration counts and runtime.

Step-size and convergence diagnostics: Beyond Armijo line search, can you provide guidance or defaults for step-size selection and stopping criteria on the manifold (e.g., gradient norm in the horizontal space, decrease of objective, or stationarity measures)?

Complexity breakdown and scalability: 1) Provide a detailed runtime decomposition (projection, subspace update, exponential map computations, storage updates) versus problem size $(n, m, r)$ and compare to RGD/RCG. A memory footprint analysis (bytes for $UQ$, $VQ$, $\Lambda$ vs. full factor storage) would also be useful. 2) Any preliminary results on multi-core or GPU implementations? Your subspace updates seem amenable to batching.

Extensions: 1) Could the approach be extended to block-subspace updates (updating multiple subspaces per iteration) to reduce overhead of sampling and gain more progress, while still keeping updates cheap via small blocks? 2) For PSD case, can the method handle constraints or regularization (e.g., trace/nuclear-norm surrogates or box constraints on eigenvalues) within the TSD framework?

Theory clarifications: 1) Assumption 3.2 uses a $C$-randomized norm condition. Can you provide intuition/estimates for $C$ as a function of $r$ for your decompositions (you give exact values), and discuss how this affects the effective step-size and convergence rate? 2) Are there conditions under which your non-smooth subspace selection yields strictly better constants than smooth selections (e.g., fewer subspaces, better alignment with geometry)?

---

### Official Review · Reviewer_ZcN1 · 2025-11-01

**Soundness:** 3
**Presentation:** 2
**Contribution:** 2
**Rating:** 4
**Confidence:** 3

**Summary:**

This paper presents a randomized subspace methods tailored for optimization on fixed-rank matrix manifolds. The authors leverage a quotient geometry framework, mapping optimization problems on fixed-rank matrices to convenient Riemannian product manifolds, enabling efficient subspace updates that avoid expensive full decompositions. They establish a non-smooth subspace selection mechanism, distinct from prior smooth selection rules in Tangent Subspace Descent (TSD) or coordinate descent literature. The paper provides mathematical analysis of the algorithm and demonstrates, through experiments on trace regression problems, the advantages of the proposed approach over standard Riemannian gradient descent.

**Strengths:**

- The lifting of optimization over fixed-rank (and PSD) matrix manifolds to a Riemannian product manifold via a quotient
structure is accurately developed, yielding access to efficient, closed-form expressions for the exponential map
and gradient projections. This structural insight forms the technical backbone of the proposed method.

-  The adaptation of the TSD framework to the horizontal decomposition on quotient manifolds is mathematically justified. Explicit convergence bounds and rate guarantees are derived under Assumption 3.2 and Theorem 3.3.

**Weaknesses:**

- The motivation behind the proposed method is not clear to me. For example, while the authors cite existing splitting-based methods, a quantitative or qualitative comparison with these relevant works is absent, making it difficult to assess the proposed method's unique contribution and advantages. Moreover, there exist established successful retraction-based methods, which offer a computationally efficient alternative. The manuscript would be strengthened by a direct comparison with such techniques.

- The experiments cover only a narrow set of cases, making it hard to assess the method’s general performance. No real datasets,
broader applications (e.g., matrix/tensor completion, low-rank approximation on practical data), or comparisons to a wider range of baseline methods.

- Regarding the presentation, the paper, especially in Sections 4 and 5, is textually dense with heavy algebra and few visual aids beyond Figure 1. Algorithm 1 is presented inline without substantial commentary or annotation. Moreover, there are multiple misprints. For example, carefully check the formulation Lemma C.1.

**Questions:**

- How does the method behave numerically when the matrix $P$ in the quotient representation becomes ill-conditioned?

- What happens in practice as the rank $r$ and dimensions $m,n$ increase? At what parameter regimes does the complexity advantage of the randomized subspace approach over standard gradient methods diminish or disappear?

- Could you clarify why is a non-smooth subspace selection better than a smooth one?

---

### Official Review · Reviewer_naKW · 2025-11-03

**Soundness:** 2
**Presentation:** 2
**Contribution:** 2
**Rating:** 4
**Confidence:** 3

**Summary:**

The paper studies the tangent subspace descent method for optimization on the fixed-rank matrix manifolds with focus on the orthogonal decomposition of horizontal space enabling random tangent subspace selection and their efficient computation. Since the underlying product manifold is a Riemannian quotient manifold, the tangent vector for gradient descent is taken from the horizontal space for higher efficiency. For coordinate-like descent, the horizontal space is partitioned orthogonally. It is shown that the coordinate-like update can be done efficiently. Similar results are extended to manifolds of low-rank PSD matrices. Experiments are conducted on synthetic datasets.

**Strengths:**

1) Take gradient vectors from horizontal space for optimization, which is efficient compared to that from the entire tangent space
2) Partition the horizontal space orthogonally for random tangent subspace selection
3) Efficient update for each random tangent subspace descent step
4) Detailed derivation of all the results

**Weaknesses:**

1. This seems a follow-up work of "Coordinate Descent without Coordinates: Tangent Subspace Descent on Riemannian" which also considered the case of product manifolds. The contribution here is on the special case of a Riemannian quotient manifold, i.e., fixed-rank matrix manifolds, which looks incremental
2. The horizontal spaces of the considered product manifolds seem already known.
3. Experiments are too insufficient to demonstrate the practical value.
4. The case of manifolds of fixed-rank PSD matrices is straightforward, given the case of fixed-rank matrix manifolds. The space could be used for more experiments.

**Questions:**

1. In experiments, only objective suboptimality is reported. what about point distance suboptimality?
2. In Figure 1(g), TSD doesn't seem to be the best. Why?

---

### Meta-Review · Area_Chair_tArX · 2025-12-23

**Summary:**

The reviewers raised many questions including the following ones, and addressing which will be necessary to clarify the contributions of the paper: (i) this is a follow-up work on an existing work about coordinate descent method for Riemannian optimization, and the contribution is incremental. (ii) the numerical experiments are limited to synthetic problems. The authors didn’t submit a rebuttal to address the issues raised in the reviews.

**Reviewer Concerns:**

The authors didn't submit a rebuttal.

**Reviewer Scores:**

The authors didn't submit a rebuttal.

---

### Decision · Program_Chairs · 2026-01-26

Reject